# Data-driven predictions of complex organic mixture permeation in polymer membranes

Young Joo Lee[1], Lihua Chen [2], Janhavi Nistane[2], Hye Youn Jang[1], Dylan J. Weber [1], Joseph K. Scott[1], Neel D. Rangnekar[3], Bennett D. Marshall[3], Wenjun Li[3], J. R. Johnson[3], Nicholas C. Bruno[4], M. G. Finn [4], Rampi Ramprasad [2] ✉ & Ryan P. Lively [1] ✉

Membrane-based organic solvent separations are rapidly emerging as a promising class of technologies for enhancing the energy efficiency of existing separation and purification systems. Polymeric membranes have shown promise in the fractionation or splitting of complex mixtures of organic molecules such as crude oil. Determining the separation performance of a polymer membrane when challenged with a complex mixture has thus far occurred in an ad hoc manner, and methods to predict the performance based on mixture composition and polymer chemistry are unavailable. Here, we combine physics-informed machine learning algorithms (ML) and mass transport simulations to create an integrated predictive model for the separation of complex mixtures containing up to 400 components via any arbitrary linear polymer membrane. We experimentally demonstrate the effectiveness of the model by predicting the separation of two crude oils within 6-7% of the measurements. Integration of ML predictors of diffusion and sorption properties of molecules with transport simulators enables for the rapid screening of polymer membranes prior to physical experimentation for the separation of complex liquid mixtures.

Membrane-based separations of nonaqueous or organic-water liquid mixtures have been identified as a key enabler for reduced carbon emissions in chemical, biochemical, and petrochemical manufacturing processes[1]. Beyond carbon emissions, membrane separation techniques can accomplish separations that are inaccessible via incumbent technologies that often rely on high temperatures to achieve a separation (e.g., separation of bio-oils and separations in other biorefineries)[2,3]. Polymer membranes based on glassy or linear polymer chemistries have been shown to be effective at a variety of liquid phase separation problems. Recently, Thompson et al., and Bruno et al. have shown that spirocyclic-based polymer membranes were capable of removing small, aromatic-rich components from crude oils[4,5].

Additionally, Chisca et al. and Li et al. have recently highlighted the ability of crosslinked polymer networks to separate light and heavy crude oils with good fluxes and selectivity[6,7]. This type of fractionation (or splitting) is attractive when hybridized with existing separation technologies (e.g., distillation) for hydrocarbon separations, and has the potential to enable the fractionation of biobased complex mixtures such as biocrude oils, crude tall oils, and others.

A critical gap for this class of membrane separations is the difficulty associated with estimating membrane performance when challenged with a given complex mixture. Current approaches rely on laborious, specialized, and Edisonian experiments that require sophisticated and time-intensive analysis to understand the

[1]School of Chemical and Biomolecular Engineering, Georgia Institute of Technology, Atlanta, GA 30332, USA. [2]School of Materials Science and Engineering, Georgia Institute of Technology, Atlanta, GA 30332, USA. [3]ExxonMobil Technology and Engineering Company, Annandale, NJ 08801, USA. [4]School of Chemistry and Biochemistry, Georgia Institute of Technology, Atlanta, GA 30332, USA. ✉e-mail: rampi.ramprasad@mse.gatech.edu; ryan.lively@chbe.gatech.edu

effectiveness of the membrane. The application of data-driven techniques has rapidly gained attraction in a wide range of research and development fields such that sophisticated machine learning (ML) techniques are now becoming omnipresent in many fields of science and engineering. In the field of materials science and development, materials informatics has blossomed in the past few years due to the increasing availability of large amounts of chemistry-processing-property data, which has promoted accelerated materials innovation. For instance, Polymer Genome (http://www.polymergenome.org) – a web-based platform, provides publicly-accessible predictive ML models for gas permeability/selectivity, solvent solubility, plus dozens of other properties relevant for a plethora of applications[8–12]. The ML models employ a state-of-the-art fingerprinting method that captures the polymer chemistry and converts it into mathematical machine-readable form, which are then mapped to properties using a variety of learning algorithms[10,13]. Such predictive capabilities can be further used to direct and guide the design of new polymer structures that meet target performance (e.g., permeability and selectivity) and property needs[14].

ML approaches have been recently applied to organic solvent nanofiltration (OSN, also referred to as solvent resistant nanofiltration, SRNF) to predict permeance of a single solvent and the rejection of a single solute[15,16]. Other methods have focused on the construction of a ML model based on guest permeances assuming a pressure-driven transport mechanism[17]. These models worked well for nanofiltration-style separations involving a solvent (or solvents) and large solutes (e.g., styrene oligomers). However, these models ultimately are incapable of predicting the separation of small molecule mixtures that are often found in practical applications. These mixtures often have numerous components concentrated almost equally such that no clear "solvent" is identifiable. Moreover, existing ML and modeling approaches are not generalizable to the sheer variety of solvent molecules nor a wide range of polymer materials under the large phase space of operating conditions (e.g., feed concentrations, pressures), which is critical for predicting the performance of a membrane[18–20]. These models are also largely focused on nanofiltration membranes, and do not accurately describe organic solvent reverse osmosis (OSRO) membrane performance.

One observation when attempting to predict the permeation of many different compounds through an OSRO membrane is that the notion of a "permeability" (i.e., the flux normalized by the membrane thickness and driving force) is not particularly useful. These permeability coefficients are often derived from single-component permeation experiments, which are not representative of the complex fluids found in multi-component separations. In complex mixture cases, the coupling of driving forces and transport parameters renders the single component permeability difficult to utilize.

In this work, we develop transport parameter predictors using ML algorithms trained on numerous (>2000 data points) experimental diffusion coefficients and sorption uptakes of organic solvents in polymers. These transport parameters were then deployed in a mass transport model based on thermodynamic driving forces. This integration of parameter prediction and transport modeling enables rapid estimation of the separation performance of complex organic liquid mixtures using any linear polymer membrane based solely on the chemical structures of the polymer and molecules to be separated as the inputs. We validated this prediction strategy with complex hydrocarbon mixtures, two crude oil mixtures, and a biofuel-type mixture.

## Results

### Design of a predictive framework for predicting any arbitrary complex mixture via any arbitrary linear polymer membrane

To enable rapid and quantitative predictions of the separation of any arbitrary complex mixture using any arbitrary linear polymer membrane, we designed a new framework where data-driven estimation of guest molecule transport and interaction parameters (e.g., diffusivity, solubility) are coupled with an $N$-component transport model based on the Maxwell-Stefan equations for polymer systems (Fig. 1). Discussion on the development of the data-driven transport modeling is described by Eqs. (1) to (13) in the Methods section. For this multi-part framework, the prediction begins by extracting the diffusion and sorption properties of single components in the complex mixture in the membrane via data-driven predictors (i.e., using the Polymer Genome approach) based on the molecular structures of the polymer and solvents. These structure-oriented properties are subjected to thermodynamic corrections with the Flory-Huggins sorption model, thus generating input transport parameters (e.g., Maxwell-Stefan diffusivities, Flory-Huggins interaction parameters) as described by Eqs. (6), (7), and (12) in the "Methods" section[21–26]. These parameters are utilized in the transport modeling. The transport model also uses as inputs the membrane separation operating conditions (e.g., transmembrane pressure, system temperature) and solvent parameters (e.g., molar volumes, vapor pressures, Hansen solubility parameters) that are necessary for thermodynamic quantification of the fugacities and the driving forces (e.g., concentration gradients of the permeants across the membrane thickness) of each penetrant. The solvent parameters were found from publicly open data sources or a structure-oriented lumping framework; the latter was applied to estimate the properties of the thousands of distinct molecules in the crude oils used in this work (described in the "Methods" section)[27,28]. A complication that arises in this method is that diffusivities of guest molecules in a polymer membrane are sensitive to the level of dilation and plasticization that the polymer is experiencing[29–32]. This has been addressed by the introduction of a "cohort diffusion" concept, in which all molecules exhibit essentially a composition-averaged diffusivity throughout the membrane[33]. This assumption significantly streamlines the modeling framework while still yielding surprisingly accurate predictions, as shown later in this article. More sophisticated implementation of the diffusion data (e.g., accounting for diffusive cross coupling) can be envisioned, and these will likely improve the accuracy of the prediction at the expense of additional computational cost for the transport model.

### Development of ML algorithms capable of predicting transport parameters

To power this predictive framework, we developed two ML models to predict the diffusion and sorption of a pure solvent into a polymer (Fig. 2). The ML models are based on a fingerprinting technique designed to interpret chemical structures of both the solvents and the polymer. These models leverage a large database of experimental guest diffusivities and solubilities that we curated for this work. This database spans a large chemical space of solvents and polymers and is available in the "Methods" section.

The models were trained on a database of experimentally-measured Fickian diffusion coefficients ($cm^2 s^{-1}$) and sorption uptakes (i.e., mass or volume fraction of solvent in polymer at some solvent activity) (Supplementary Fig. 1). For simplicity, we use a molar volume ($cm^3 mol^{-1}$) to represent the size of the solvent molecules while acknowledging that other size-based descriptors exist. Existing data for the diffusion and sorption of molecules larger than $250 cm^3 mol^{-1}$ is exceedingly rare in the literature. This lack of data has important implications for complex molecules containing large solutes; for instance, the crude oils used in this work contain molecules with molar volumes >$1000 cm^3 mol^{-1}$. There is thus a concern that the working range of the model will be limited to relatively small molecules (i.e., <$250 cm^3 mol^{-1}$), while the predictions for larger molecules will be in an extreme extrapolative regime with large uncertainties. Moreover, molecules such as these are sufficiently large that experimental determinations of diffusivity are intractable due to slow uptake kinetics.

## Data-driven prediction of complex mixture permeation

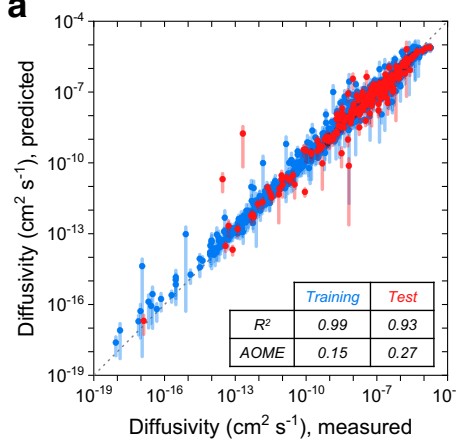

**Fig. 1 | Schematic diagram of data-driven transport modeling framework.**
Polymer structures and solvent mixtures are converted to simplified molecular-input line entry system (SMILES) strings and used as inputs for machine-learning algorithms designed to relate polymer-solvent structure to solvent diffusivities ($D$) and solubilities ($S$) within polymer membranes. These parameters – in addition to the various physicochemical properties of the solvents (e.g., molar volumes ($\hat{V}$), vapor pressures ($p^{sat}$), Hansen solubility parameters ($\delta$)) at the desired operating conditions (e.g., pressure ($P$), temperature ($T$), composition of the feed mixture ($x^f$), membrane thickness ($\ell$)) – are then used as inputs into an N-component Maxwell-Stefan model that outputs a vector of fluxes ($N$) and compositions ($x^p$) for each component permeating through the membrane.

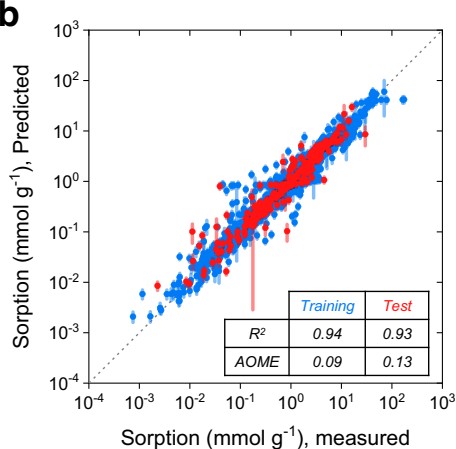

**Fig. 2 | Development of machine learning (ML) diffusion and sorption parameter prediction models. a, b** Parity plots between experimentally obtained and ML predicted diffusion coefficients and sorption uptakes, respectively (the methods of the model development are described in "Methods" section). Both diffusion and sorption models are trained using 10-fold cross-validation (CV). The 10 models from the 10 CV splits were used to make predictions on the 90% training (blue) and the 10% test set (red). The error bars on each point represent the standard deviations from 10 predictions. $R^2$ and AOME in the plots are defined as the coefficient of determination and averaged order of magnitude error, respectively (Eq. (14)). Source data are provided in the Source Data file.

To enable reasonable predictions for molecules that deviate in size from the common solvents found in the dataset, we extracted a correlation between the transport parameters and the size of the solvents and encoded this correlation in our neural network so that extrapolation to large molecular sizes is guided by expected physical principles. Generally, the diffusion coefficient of organic solvents in polymers decreases with increasing size of the guest molecules. This trend is clearly supported by previous experimental reports that demonstrated a molar volume scaling based on the class of molecule diffusing (i.e., aromatic, aliphatic, etc.)[34,35]. Based on this, we

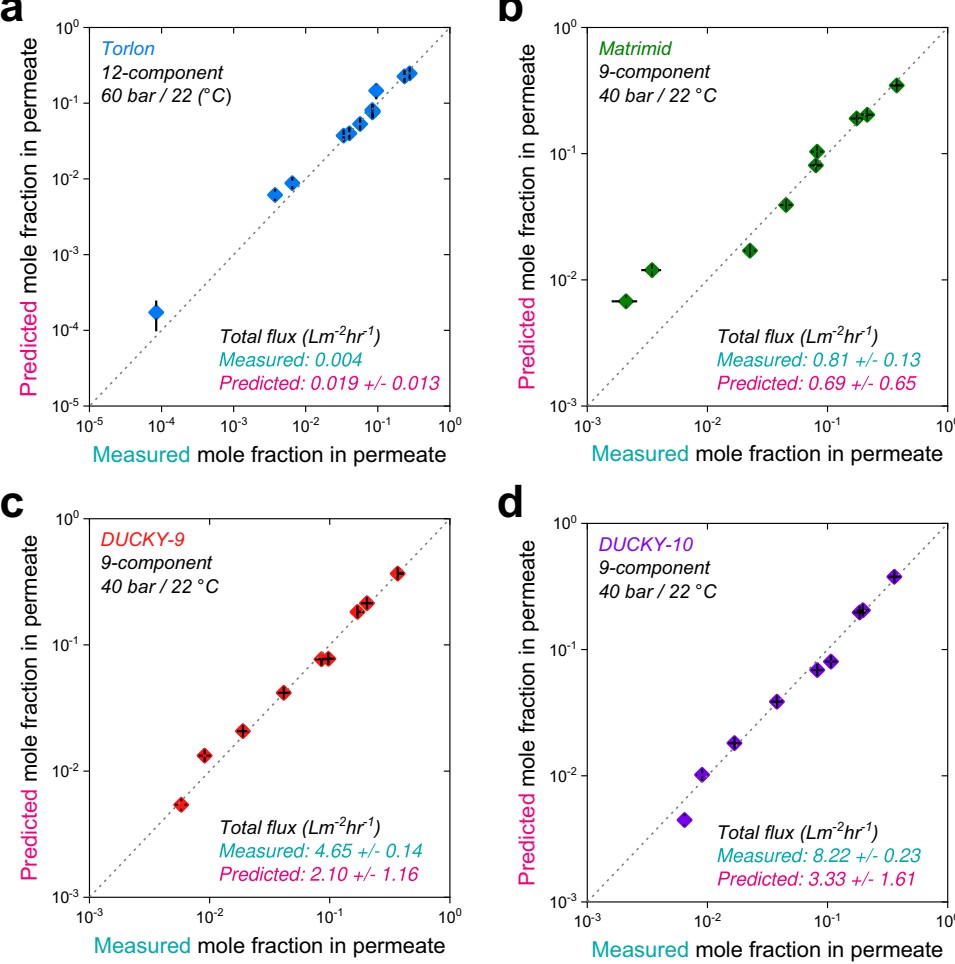

**Fig. 3 | Evaluation of data-driven transport model for hydrocarbon mixture separations.** (**a**-Torlon, **b**-Matrimid, **c**-DUCKY-9, **d**-DUCKY-10) Comparisons between experimentally-measured permeate concentrations and predictions. All experiments and predictions were performed at 22 °C. The pressures in the plots indicate the transmembrane pressure (i.e., the applied pressure at upstream side with an atmospheric pressure at downstream). The error bars represent the standard deviation of the permeate concentration predictions for each molecule, and the deviations are from uncertainty in the machine learning (ML) sorption model parameter predictions. The deviations in the total flux predictions are from the uncertainty in the ML diffusion model predictions (Supplementary Tables 3–6). Source data are provided in the Source Data file.

incorporated a power law scaling between the guest molar volume and diffusivity through the polymer host into our ML architecture based on neural network (Supplementary Fig. 2 and Supplementary Table 1). Learning curves compare errors in training sets and test sets after infusion of the physics into the models (Supplementary Fig. 3). The diffusion model trained in such a way exhibited lower errors compared to the model without physics, suggesting that the molar volume scaling of the diffusivity infuses sufficient physics to extract reasonable diffusivities of larger molecules not seen in the database.

The sorption and diffusion coefficient estimates are then fed into a Maxwell-Stefan $N$-component mass transport model developed for swollen polymer membranes. Additional inputs to this transport model include the feed mixture composition, transmembrane pressure, membrane thickness, system temperature, and Hansen solubility parameters. Using this information, a 2-point boundary value problem (i.e., a boundary at the upstream and downstream sides of the membrane) must be solved for the $N$-component system, which includes the polymer membrane and all of the solvents in the mixture. While it is true that the ordinary differential equation (ODE) solver only requires one initial boundary condition at the upstream side of the membrane, the unknown variables of the total flux and permeate composition make the downstream boundary condition an unknown as well. This

necessitates an iterative numerical procedure, thus creating the 2-point boundary value problem (see "Methods" section for further details). Lastly, the model equations have been transformed to deal with large numbers of components in the complex mixtures (as large as 400 components in this work) such that the model was found to converge to a solution within 24 h.

## Validation of the data-driven prediction model via complex mixture separations

Several complex mixture separation experiments were conducted to evaluate the predictions from the new data-driven transport model (Fig. 3). We performed the separation of 9 and 12 component hydrocarbon mixtures in the following uncrosslinked polymer membranes: Torlon® 4000T-LV (a commercial polyamide-imide), Matrimid® 5218 (a commercial polyimide), and DUCKY-9 and DUCKY-10 (spirocyclic polytriazoles for which diffusion and sorption data does not yet exist) (Supplementary Fig. 4, chemical structures of the polymers studied). The hydrocarbon mixtures consisted of 9 to 12 components of nonpolar hydrocarbons with various sizes and classes (e.g., aromatic, aliphatic) at high concentrations (Supplementary Table 2). We parameterized the transport model with diffusivities and solubilities for liquid phase solvents at ambient conditions that were predicted via

the ML models (Supplementary Tables 3–6). To calculate the partial and total flux from each separation, the thickness of the membrane selective layer was visually determined from cross-sectional SEM images of the membranes (Supplementary Fig. 5). In general, the prediction framework accurately describes the complex mixture separations, although we observe some deviations between the experimental measurements and the model predictions of the permeate composition and fluxes for the commercial polymers (Fig. 3a, b). The average deviations in the Torlon and Matrimid membranes were 7.3% (0.09 order of magnitude deviated) and 11.0% (0.16 order of magnitude deviated) each based on root mean square percentage error, RMSPE, and averaged order of magnitude error, AOME (Eqs. (14) and (15), and Supplementary Tables 7 and 8). Importantly, the total fluxes are reasonably predicted (within 0.07–0.67 order of magnitude of the measurements). In general, the model shows more accurate predictions for smaller molecules in the feed mixture, which is reasonable as this coincides with the majority of the data in the database. We contrasted the commercial polymers (for which diffusion and sorption data exist) with experiment-model comparisons for DUCKY-9 and DUCKY-10 membranes (for which diffusion and sorption data do not exist). Here, we observed that the measured permeate compositions are within 4.8% (0.05 order of magnitude deviated) and 5.9% (0.06 order of magnitude deviated), respectively, of the model permeate predictions showing that most of the measured permeate concentrations are located in the 95% confidence intervals (Fig. 3c, d and Supplementary Tables 9 and 10). Since we assumed no diffusion selectivity as a result of the cohort diffusion approach utilized here, the separation must be essentially driven by sorption selectivity between molecules transporting through the membrane. The Torlon and Matrimid membranes likely exhibit some diffusion selectivity, which is not captured in the transport model. More sophisticated estimates of the diffusion selectivity can be envisioned, but ultimately require more complete concentration-dependent diffusion datasets for a variety of polymers. As most OSRO membranes will likely operate in regimes in which the polymer is dilated (to provide meaningful solvent fluxes), we believe that the cohort diffusion approach provides a meaningful path forward for the prediction of complex mixture permeation. An important observation of the predictions made is that the data-driven approach was able to correctly order the polymers based on their respective fluxes for the given separation, even though the predicted fluxes were under-estimated when compared to the measured values. This discrepancy in the predicted and measured fluxes many have resulted from discrepancies in the diffusivity estimates, which were generally larger and more impactful than the sorption estimates used for predicting the fluxes. The datasets represent diffusivities in a range of conditions but often at unit activity for the solvents. In complex mixture separations, the individual activity of the various compounds in the membrane will not be unity. Additionally, since a mixture of solvents is present in real experiments, the polymer will be in a state of dilation that is distinct from the state of dilation in unit activity diffusion experiments where diffusivity of a pure solvent was determined. A theory related to this effect was envisioned by introducing the free volume change of a polymer membrane when exposed to a complex mixture[33]. However, this additional complexity was eschewed here due to acceptable prediction accuracy found using the significantly simpler average diffusivity approach.

To further evaluate the prediction capability of the data-driven approach beyond hydrocarbon mixtures, we conducted a separation of a binary mixture consisting of methanol and guaiacol as a representative of a biofuel-type mixture (Supplementary Fig. 7 and Supplementary Tables 11–13). The separation experiment was conducted using three DUCKY-9 membranes. As before, the ML dataset had not been previously trained with these specific solvent molecules for DUCKY-9. Notably, the separation factor of guaiacol was accurately predicted, and the flux was closely predicted to that of the experimental measurements. This outcome suggests that the data-driven permeation predictions may have broader applicability beyond simple hydrocarbons, which have been the focus of this work, although our limited set of experiments prevents further generalization of this conclusion.

## Fractionations of real crude oils by polymer membranes and validation of the data-driven predictions

Effective fractionation of highly complex mixtures such as crude oils (which typically contain >60,000 components) has been highlighted in several recent studies[4–7]. We utilized the data-driven transport model developed here to predict the separation of two whole crude oils (Permian crude oil and Arabian Light crude oil). To achieve this, ~400 of the most abundant molecules in the crude oils were chosen, and the diffusivities and solubilities of these molecules in two polymers (SBAD-1 and DUCKY-9) were generated using our ML models. The ML algorithms have been trained on a limited number of experiments for SBAD-1 (9 instances of pure component sorption and diffusion experimental measurements) while DUCKY-9 does not exist in the dataset[33]. The list of guest diffusivities and solubilities in these two polymers were used to parameterize a 400-component Maxwell-Stefan transport model with a cohort diffusion assumption. The results of these two predictions are shown in Fig. 4. Here, we compare the experimental and model-predicted permeate compositions from the membrane as well as the differential boiling point curves for the crude oil permeates. The prediction on mole-based permeate concentration of Permian crude oil fractionated by SBAD-1 membrane gives a generally good agreement with an overall error range of 6.8% for all molecules (0.20 order of magnitude deviated; Fig. 4a and Supplementary Table 14). We note that predictions of the permeate concentration for the higher boiling point molecules deviated more from the experiments than the lower boiling point molecules. This mismatch for the higher boiling point components is likely a result of the data scarcity for diffusivity/solubility ML model training and potentially from difficulties in detecting these compounds in the experimental work-up of the crude oil permeate. The physics-guided ML parameter estimation enables reasonable estimates of the partial fluxes of these molecules. The fractionated stream could be potentially used as sources of gasoline and kerosine, which are products boiled below 200 °C; these light boilers were enriched up to ~57% at the permeate stream from 40% at the feed stream (Fig. 4b). We predict 49% of the permeate is composed of these light ends, which is close to but somewhat less than the measured result. Moreover, the SBAD-1 membrane rejected relatively higher boiling point molecules (>400 °C) down to a total content of ~7% from a feed of 21%. The data-driven transport model slightly under-predicted the rejection of these high boiling fraction (10% content in the permeate). Interestingly, the model predicts $0.88 \pm 0.43 \, \text{L} \, \text{m}^{-2} \, \text{h}^{-1}$ for the Permian crude oil flux through SBAD-1, which is close to the measured flux of $0.85 \pm 0.35 \, \text{L} \, \text{m}^{-2} \, \text{h}^{-1}$. Despite the generally good agreement between the model predictions and experiments for both the separation performance and the fluxes, there are some noteworthy discrepancies between the experiments and predictions. First, diffusion and sorption processes are dependent on the system temperature. The diffusivities and solubilities in this permeation prediction were deduced from the temperature range from 25 °C to 40 °C, which is the temperature range of our dataset, while the tested temperature was 130 °C. Recent work on the thermodynamics of crude oil permeation highlight that the temperature effect has only moderate effects on the fluxes of individual molecules[36]. The relatively moderate temperature dependencies observed in the hydrocarbon permeation systems here may not necessarily apply to other systems, which is a current shortcoming of the model. To address this lack of built-in temperature dependence in the model, it will likely be beneficial to develop separate predictors for estimating the activation energy of diffusion and the heat of sorption

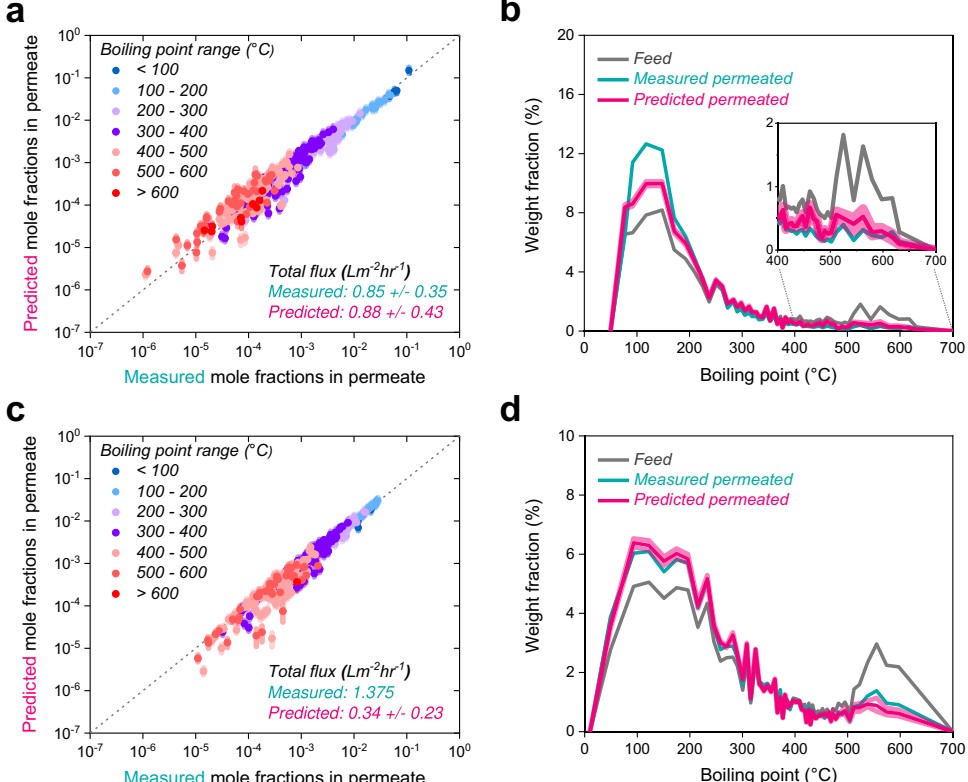

**Fig. 4 | Predictions of crude oil fractionation compared to experiments.**
**a, c** Parity plots comparing experimental and predicted permeate mole fractions after the fractionation of Permain crude oil by SBAD-1 membrane (**a**) and Arab Light crude oil by DUCKY-9 membrane (**c**). Blue-to-red colors are assigned to different boiling point ranges of molecules in the crude oil mixtures. The shaded area around each point represents the standard deviation of the permeate concentration predictions for each molecule. The deviations are from the uncertainty in the machine learning (ML) sorption model parameter predictions. The deviations in the total flux predictions are the uncertainty in the ML diffusion model predictions.
**b, d** Differential weight fraction relative to boiling points of molecules in the Permian crude oil mixture before and after fractionation by SBAD-1 membrane (**b**) and in the Arab Light crude oil mixture before and after fractionation by DUCKY-9 membrane (**d**). The curve shows the local slope of the concentration/boiling point over a period of 6 molecules. The lighter shade displays the deviation in the predicted weight fractions of the permeate. Source data are provided in the Source Data file.

for individual guest molecules and incorporating these parameters into the transport model. Estimation of these parameters can potentially be achieved through data-based methods and simulations such as molecular dynamics.

Another source of error in the model is the cohort diffusion assumption. We note that magnitude of the discrepancies between the model and the experiments are most significant for the components with the highest and lowest fluxes. While the cohort diffusion assumption implies that all penetrants have the same diffusivity within the membrane, in reality the diffusivities are likely to exist on a distribution such that the penetrants farthest away from the mean diffusivity exhibit the largest deviations from the experimental observations. Additionally, in the presence of dissimilar molecules with low chemical affinity, the cohort diffusion assumption might be deemed invalid.

Compared to the Permian crude oil fractionation via SBAD-1, we observed better agreement between the permeate composition prediction and the experiment in the Arabian crude oil fractionation via DUCKY-9 membranes (Fig. 4c, d). We observe generally good agreement between the experiments and models for the entire boiling point range of the Arab Light crude oil (Supplementary Table 15). In this experiment, the Arab Light feed was comprised of ~26% light boilers (<200 °C) and 31% higher boiling point molecules (>400 °C). The separation was taken to a 30% stage cut. Analysis showed that the permeate was comprised of 32% light boilers and the prediction estimated this value to be ~33%. Furthermore, the high boiler content was reduced to 21% in the permeate and was predicted to be 19%. The flux

in the DUCKY-9 membrane is under-predicted ($0.34 \pm 0.23$ L m$^{-2}$ h$^{-1}$), compared to the measured flux of 1.375 L m$^{-2}$ h$^{-1}$. The higher flux measured here could be attributed to the batch-type fractionation test with a high stage-cut (e.g., 30%), where the feed concentration may have become polarized during the test. To account for the time-related concentration change in feed during batch-type fractionation systems, an additional step that considers the concentration change over time or stage-cut could potentially be included in the transport modeling. Additionally, it is important to note that very large molecules (e.g., high boilers in crude oil) likely have reduced diffusivities relative to the average diffusivity exhibited by the smaller molecules. however, it is still promising that the error range is within the same order of magnitude considering the significant complexity of this permeation system.

## Discussion

Integration of ML property predictors relevant for molecular sorption and diffusion through polymers with multi-component transport models provides a powerful and facile method for the estimation of complex mixture permeation through membrane materials. The inputs into the model are generally accessible: chemical structure of the polymer and solvents, estimates of the feed composition and solvent properties and approximations of the membrane geometry (i.e., membrane thickness). We have demonstrated the utility of this approach for complex hydrocarbon mixture separations and a binary mixture separation of oxygenated molecules in several glassy polymers. Accurate estimates of a solvent's solubility and diffusivity within

a polymer are necessary to enable predictions of solvent permeation through polymer membranes. These two parameters depend on the current state of the polymer, including its level of dilation, aging, and processing history, as well as the mixture that permeates through the membrane. Although our current model can estimate membrane performance with surprising accuracy without taking into account these complicated issues, future models that incorporate the physics of the polymer and its free volume into the transport framework should result in improved model accuracy.

## Methods

### Materials

Commercially available Torlon® 4000T-LV and Matrimid®5218 were purchased from Solvay Advanced Polymers (Alpharetta, GA) and Huntsman, respectively. SBAD-1, DUCKY-9, and DUCKY-10 polymers were synthesized from literature procedures detailed in two previous publications[4,5]. Two real crude oils (Permian and Arabian light crude oils) were provided from ExxonMobil Technology and Engineering Company. The experimental results of the two fractionations were published previously[4,5]. All other chemicals (p-xylylene diamine, lithium nitrate, chloroform, tetrahydrofuran, 1-methylpyrroldone, ethanol, methanol, hexane, toluene, toluene, Tert-butyl benzene, 1,3,5-triisopropyl benzene, n-Octane, iso-Octane, iso-cetane, methylcyclohexane, decalin, 1-methylnaphthalene, o-xylene, propyl benzene, mesitylene, n-butylcyclohexane, tetralin, bi-phenyl, dodecylbenzene, 1,3,5-triphenylbenzene, 1,3,5-Tris[(3-methylphenyl) phenylamino]benzene), methanol, and guaiacol were purchased from Sigma Aldrich, Alfa Aesar, or TCI and used as received.

### Membrane fabrication

In this work, two different forms of asymmetric membranes were fabricated: hollow fiber membranes and thin-film composites. Defect-free Torlon® hollow fiber membranes were fabricated by the spinning procedures established previously[18]. Specifically, the polymer dope composed of 34, 47.2, 11.8, and 7 wt% of polymer, 1-methylpyrrolidone (NMP), tetrahydrofuran (THF), and ethanol, respectively. For the bore fluid, 80 wt% of NMP was diluted with 20 wt% of deionized water. Note that the Torlon® 4000T-LV was dried under vacuum at 110 °C overnight and then used. The temperature of quench bath was 50 °C, and the dope was degassed at 60 °C. The extruded fibers were immersed in the quench bath after passing through a 0.23 m air gap. The flow rates of the dope and bore fluid were 180 and 60 mL per hour. The spun hollow fiber was taken up at a rate of 32 m per minute. After spinning, the membranes were soaked in deionized water (3 days, changing each day), methanol (3 h, changing each hour), and hexane (3 h, changing each hour) sequentially to remove residual solvents. The fibers were dried under ambient air for an hour and then dried under vacuum at 120 °C for 12 h.

To fabricate the thin-film composite (TFC) membranes, cross-linked Matrimid supports were first made using the following procedures. Matrimid®5218 and lithium nitrate (LiNO$_3$, a pore-former), were dried under vacuum at 110 °C. A dope for the support was prepared with Matrimid®5218 (16 wt%), LiNO$_3$ (3 wt%), NMP (69 wt%), THF (10 wt%), ethanol (1 wt%), and deionized water (1 wt%). The dope was homogeneously mixed on a roller for at least a day and degassed for 2 h before casting. The dope was cast on a glass plate with a 10 MIL casting blade, and the cast dope was transferred to deionized water bath to be solidified by non-solvent driven phase inversion. The resulting sheet was soaked in deionized water for 3 days and further immersed in methanol and hexane three times each (1 h per round) to remove residual solvents and salts. After drying for 1 h in ambient air, the flat sheets were cut into circular coupons with an effective area of 10.25 cm². The circular supports were immersed in a cross-liking solution (5 g of p-xylene diamine in 100 mL methanol). The immersion

was performed for 24 h, and then the same solvent exchange procedures were conducted to remove residual cross-linkers.

To fabricate the TFC membranes of Matrimid®, DUCKY-9, and DUCKY-10, the polymers were dissolved in anhydrous chloroform. Here, the concentration of polymer in the chloroform-based dope was 1 wt% and the polymers were dried under vacuum at 110 °C. The prepared dopes were chilled in a fridge set in 5 °C before being used. Finally, the TFC membranes were fabricated by spin-coating method where the skin layer is formed on the top of the support. For spin coating, 0.5–0.7 mL of each polymer dope was dropped on a plate in the spin-coater with a rotating speed of 1200 rpm. Tests of crude oil fractionations (Permian crude oil via SBAD-1 membrane and Arabian crude oil via DUCKY-9 membrane) were performed previously; the results were used in this work[4,5]. To make the membranes used in the crude oil fractionation tests, chloroform solutions with SBAD-1 (2 wt%) and DUCKY-9 (1 wt%) were blade-casted on a cross-linked polyetherimide (PEI, ULTEM 1000) support and the films were dried overnight at room temperature in a fume hood before circular coupons with an effective area of 14.6 cm² were cut out for testing.

### Membrane testing

A 12-component hydrocarbon mixture consisting of aromatics with various sizes and boiling points (Table S2) was prepared to test the separation of defect-free Torlon® hollow fiber membranes. The mixture permeation test was performed using a home-built high-pressure syringe pump (500D, Teledyne Isco) at 295 K[18]. The applied pressure was ramped at around 1 bar per second until the desired pressure of 60 bar was achieved. The permeate was collected at stage cuts of around 20 wt% (stage cut is the mass fraction of the feed that permeates through the membrane). The concentration of the permeate was analyzed by gas chromatographic methods (7890B GC, Agilent) and the amount of the permeate was normalized by the effective area (200 cm²) and sample collection time to ultimately obtain partial fluxes of every molecule in the mixture.

A 9-component hydrocarbon mixture was also prepared as a "synthetic" crude oil and tested using TFC membranes with Matrmid®, DUCKY-9, or DUCKY-10 selective layers (Table S2). Permeation was measured with a custom-built cross-flow system pressurized up to 40 bar at upstream side by an HPLC pump (Azura P 4.1S, Knauer) at 295 K[5,37]. The permeation experiments were conducted for at least 48 h to ensure steady-state flux. The concentration of the permeate was analyzed by gas chromatographic method (7890B GC, Agilent) and the amount of the permeate was normalized by the effective area (10.25 cm²) and time to obtain the partial fluxes of every molecule in the mixture.

The tests of crude oil fractionations were conducted in prior work[4,5]. Briefly, batch-type separation with 49 mm diameter coupons of SBAD-1 and DUCKY-9 were loaded into a Sterlitech HP4750X stirred dead-end-cell. The cell was initially loaded with 50 g of toluene, which was allowed to permeate overnight at room temperature and 55 bar N$_2$ head pressure. The cell was then depressurized and loaded with 100 g of whole crude oils and 55 bar N$_2$ head pressure was again applied. The cell was stirred at a constant rate of 400 rpm. A cold trap cooled by dry ice was set up to collect the permeate to prevent loss of the light ends. The temperature of the cell was slowly increased up to 130 °C until permeate flow was observed. After sufficient permeate had been collected, the cell was cooled and depressurized. The permeate and feed samples were analyzed using simulated distillation (SIMDIS) and 2-dimensional gas chromatography (GCxGC).

### Crude oil molecular compositions and properties calculation

The detailed molecular compositions of crude oils are developed based on a structure oriented lumping (SOL) framework[27,28]. SOL is a

mathematical group representation of petroleum molecules, which is robust for representing crude complex mixtures, calculating molecular and mixture properties, creating reaction networks, and developing process models. The SOL-based compositional models of crude oils are constructed through large scale analytical characterizations and extensive modeling effort to observe that hundreds of thousands of organic molecules exist in the mixtures. Many property models have been developed in the SOL framework, including molecular density, boiling point, vapor pressure, and Hansen solubility parameters; these property models were specifically applied in this work. These property models were mainly developed by empirical correlations or group contribution methods based on literature and internally measured property values. The details of SOL modeling framework to derive crude oil compositions as well as their properties calculations are from ExxonMobil Technology and Engineering Company proprietary technologies.

### Scanning electron microscopy

Scanning electron microscopy (SEM) was used to obtain high resolution images of the membranes (Hitachi SU8010). After testing, a small portion of each membrane was soaked in hexane for 10 min and then immersed in liquid nitrogen. After frozen, the pieces were broken with the broken edge facing upward in the sample plate of the SEM. To estimate the thickness of the membranes, the cross-sectional images of the membranes were used. Prior to imaging, the samples were sputtered with gold (Quorum Q-150T ES). Images were obtained with a voltage of 3 kV and 5 kV, and a current of 10 μA.

### Composition analysis of liquid mixtures

The composition of permeate and feed samples (12-component and 9-component hydrocarbon mixtures in this study) were determined by gas chromatography (7890B GC, Agilent). This work contains two experimental results of two crude oil fractionations. The compositions of the feed crude oils and the permeated crude oils were analyzed by models based on a structure-oriented lumping approach constructed through extensive experiments and modeling efforts to acquire the concentrations of thousands of molecules within the crude oils. The details of this process are proprietary to ExxonMobil Technology and Engineering Company.

### Transport modeling for solution-diffusion permeation

We have previously described the development of a Maxwell-Stefan framework for solution-diffusion permeation[33]. This Maxwell-Stefan framework can be used to predict the flux of each component in a complex mixture. Note that methods based on Fick's first law with "frame of reference" corrections also exist and are potentially workable for this problem. However, they are more complex to solve. Therefore, we choose to utilize the Maxwell-Stefan formula as it is straightforward to deploy for highly complex mixtures with many components. The main framework that we have used in this work is as follows (for $i = 1, 2, \ldots, n$)[7]:

$$(N^V) = -[B]^{-1}[\Gamma]\frac{d\phi_{1:n}^m}{dz} \quad (1)$$

$$[B]_{ii} = \sum_{j=1; j\neq i}^{j=n} \frac{\phi_j^m}{\text{Đ}_{ij}^{v,m}} + \frac{\phi_{n+1}^m}{\text{Đ}_{i,n+1}^{v,m}}; \quad [B]_{ij, i\neq j} = -\frac{\phi_i^m}{\text{Đ}_{ij}^{v,m}} \quad (2)$$

$$\Gamma_{ij} = \frac{\phi_j^m}{f_i^m}\frac{\partial f_i^m}{\partial \phi_j^m} = \phi_i^m\frac{\partial \ln a_i^m}{\partial \phi_j^m} \quad (3)$$

Here, there are $n$ components permeating through the membrane. Then, the $(n+1)^{st}$ component indicates the polymer membrane, z is the dimension across the membrane thickness, $(N_i^v)$ is an $n$-dimensional vector of fluxes (L m$^{-2}$ h$^{-1}$) of permeants, $[B]$ is an $(n \times n)$-dimensional diffusional matrix, $[\Gamma]$ is an $(n \times n)$-dimensional thermodynamic coupling matrix, $\phi^m$ is an $(n+1)$-dimensional vector of volume fraction of each guest molecule in the membrane (volume of solvent per total volume of polymer + solvent system), $\frac{d\phi_{1:n}^m}{dz}$ is an $n$-dimensional vector of the first $n$ volume fraction gradients with respect to z, $\text{Đ}_{i,n+1}^{v,m}$ is the volume-based Maxwell-Stefan diffusivity of a single component $i$ (which is thermodynamically corrected diffusivity by Eq. (12)), $\text{Đ}_{ij}^{v,m}$ is the pairwise "frictional coupling" for every molecule in the mixture permeating through the membrane (i.e., solvent-solvent or exchange diffusivities). This friction is not considered in this work, as our cohort diffusion assumption leads to all molecules having the same diffusivity (this will be discussed later in Eq. (13)), $f_i^m$ is the fugacity of component $i$ sorbed in the membrane, and $a_i^m$ is the activity of component $i$ sorbed in the membrane. Solving the equations above to get $(N_i^v)$ is the main challenge in prediction of complex mixture separations, and all parameters in the equations will be parameterized by either ML predictions or standard rules of thermodynamics.

The framework has been proposed to predict the permeation in an asymmetric membrane, which consists of sequential sectors progressing in order of (i) the upstream side of feed, (ii) active layer ($z = 0-\ell$), (iii) support layer, and (iv) the permeate (downstream) side. We assume no resistance through the support layer. The numerical methods to solve the equations are also detailed in previous work[7]. Briefly, the Maxwell-Stefan equations are solved by thermodynamic rules and mass balances at the interfaces. The first assumption is the equilibrium between the bulk fluid and the mixture sorbed in the upstream membrane face at $z = 0$[7]:

$$a_{i,0}^m = a_{i,0}^{fluid,upsteam} = x_i\gamma_i \quad (4)$$

$a_{i,0}^{fluid,upsteam}$ is the activity of component $i$ in the feed fluid, $x_i$ is the mole fraction of component $i$ in the feed fluid, and $\gamma_i$ is the activity coefficient of component $i$. The activity coefficients of hydrocarbons in a 9-component mixture were calculated using the PC-SAFT thermodynamic activity coefficient model in ASPEN Plus. To apply the activity coefficient model to the transport simulation, the phase equilibrium expressions (Eqs. (4) and (9)) are updated with values of the estimated activity coefficients each iteration for the downstream phase equilibrium. The upstream equilibrium is fixed at a given temperature, pressure, and composition. The result, presented in Supplementary Tables 8–10, reveals that most of the activity coefficients are nearly unity, despite the concentrated nature of the mixture. However, it is important to note that this ideal mixture assumption may not be applicable to other complex mixtures, particularly those containing water or a combination of polar and nonpolar components. In such cases, more sophisticated thermodynamic models for both the feed and permeate phases will be necessary to accurately account for the side variation in activity coefficients.

Additionally, we assumed Flory-Huggins type sorption model for fugacity (activity) calculations[7]:

$$\ln(a_i^m) = \ln\phi_i^m + (1 - \phi_i^m) - \sum_{\substack{j=1 \\ j\neq i}}^{n+1}\frac{\bar{V}_i}{V_j}\phi_j^m + \left(\sum_{j=1}^{i-1}\chi_{ji}\phi_j^m\frac{\bar{V}_i}{V_j} + \sum_{j=i+1}^{n+1}\chi_{ij}\phi_j^m\right)\left(\sum_{\substack{j \\ j\neq i}}^{n+1}\phi_j^m\right)$$
$$- \sum_{\substack{j=1 \\ j\neq i}}^{n}\sum_{\substack{k=j+1 \\ k\neq i}}^{n+1}\chi_{jk}\frac{\bar{V}_i}{V_j}\phi_j^m\phi_k^m \quad (5)$$

$\bar{V}_i$ and $\bar{V}_m$ are partial molar volumes of guest component and membrane, respectively. In this work, the partial molar volume of each

component is assumed to be equivalent to its molar volume at pure condition, 298 K, 1 atm. To implement Eqs. (4) and (5), the volume fractions of each permeant at the upstream face of the membrane ($\phi_i^m$) are solved when the unknown parameters ($\chi$ values) are defined. Here, $\chi_{i,n+1}$ or $\chi_{j,n+1}$ is the Flory-Huggins interaction parameter between polymer and solvent. To extract this parameter, the sorption uptakes (mmol g$^{-1}$) of solvent molecules at unit activity predicted from the ML sorption model are transformed to unitless volume fraction (volume fraction of solvent in total, polymer + solvent, system, $\phi_i^m$) as follows with an assumption of a constant density of the polymer membrane:

$$\frac{mmol}{g\ polymer} \cdot molecular\ weight\ of\ solvent\left(\frac{g}{mol}\right) \cdot 1000\left(\frac{mmol}{mol}\right)$$
$$\cdot \frac{polymer\ density(\frac{g}{cc})}{solvent\ density(\frac{g}{cc})} = \frac{V_{solvent}}{V_{polymer}} = \frac{\phi_i^m}{1-\phi_i^m} \tag{6}$$

Then, the Flory-Huggins interaction parameters between the solvent and the polymer are calculated using the single-component Flory-Huggins model (Eq. (7)) at unit activity (and are assumed constant with respect to solvent concentration in the membrane):

$$\ln\left(\frac{f_i^m}{f_i^o}\right) = \ln(a_i^m) = \ln\phi_i^m + (1-\phi_i^m) - \frac{(1-\phi_i^m)\bar{V}_i}{\bar{V}_m} + \chi_{i,n+1}(1-\phi_i^m)^2 \tag{7}$$

Here $f_i^m$ is the fugacity of component $i$ in the membrane and $f_i^o$ is the fugacity at a refence state (e.g., saturation vapor pressure of pure component $i$ at a given temperature, $p_i^{sat}$).

Other $\chi$ values in Eq. (5) (e.g., $\chi_{ji}$, $\chi_{ij}$, and $\chi_{jk}$ when all $i$, $j$ and $k$ are not ($n+1$)) are the binary solvent-solvent interaction parameters that are calculated using a modified Hansen solubility theory (Eq. (8)) in which the subscript AB can apply for $ji$, $ij$, and $jk$:

$$\chi_{AB} = \frac{(\bar{V}_A\bar{V}_B)^{0.5}}{RT}\left[(\delta_{D,A}-\delta_{D,B})^2 + 0.25(\delta_{P,A}-\delta_{P,B})^2 + 0.25(\delta_{H,A}-\delta_{H,B})^2\right] \tag{8}$$

where $R$ (8.314 J mol$^{-1}$ K$^{-1}$) is the gas constant, $T$ (K) is the system temperature, and $\delta_D$, $\delta_P$, and $\delta_H$ are Hansen solubility parameters for dispersion, polarity, and hydrogen-bonding each with SI unit of MPa$^{0.5}$. This accounts for the chemical interaction between the molecules within the membrane. Solving Eq. (4) with Eqs. (5)–(8) renders the volume fractions of each permeant at the upstream face of the membrane ($\phi_{i,0}^m$). Another solubility model proposed in a previous study to describe the solubility of a solvent in a polymer consists of two distinct components: Langmuir-type filling of microvoids and Flory-Huggins swelling-type sorption[7]. However, this model requires fitting two-parameter isotherms for both the Langmuir and Flory-Huggins components. In contrast, the current study employs the Flory-Huggins and competitive Flory-Huggins models, which can be developed with only one parameter, the Flory-Huggins parameter denoted as $\chi_{i,n+1}$ in Eq. (7). While the two-parameter isotherm would be more accurate, the one-parameter Flory-Huggins models were utilized in this work to streamline the predictions of sorption uptakes at unit activity from the ML models. The Flory-Huggins model is still a useful tool for describing the sorption of organic liquids or vapors in polymer systems, even those that are glassy in nature. This is due to the fact that the sorption of organic solvents can decrease the glass transition temperature of the polymer to a point where Flory-Huggins-type sorption behavior is observed[21–25]. To improve the robustness and accuracy of the data-driven approach, it is possible to envisage the inclusion of other parameters such as the concavity/convexity of an isotherm through the use of additional ML algorithms in the future. By integrating these parameters with the existing Flory-Huggins sorption model utilized in this study, the predictive capability of the model could be improved.

Next, the fluid on the permeate-side of the active layer (i.e., $z=\ell$) is assumed to be in equilibrium with the fluid composition throughout the porous support layer. Assuming the activities of these fluids equal and that the pressure difference between upstream and downstream ($p^{upstream}-p^{downstream}$), occurs at the downstream interface (z = $\ell$):

$$a_{i,\ell}^m = a_{i,\ell}^s \exp\left[-\frac{\bar{V}_i}{RT}(p^{upstream}-p^{downstream})\right] \tag{9}$$

$a_{i,\ell}^m$ is the activity of component $i$ in the active layer and $a_{i,\ell}^s$ is the activity of component $i$ in the support layer. $\bar{V}_i$ is the partial molar volume (assumed to be the pure solvent molar volume for simplicity), $p^{upstream}$ is the upstream liquid phase pressure, and $p_i^{downstream}$ is the downstream liquid phase pressure. To obtain the driving force across the membrane, the next step is to estimate the unknown volume fraction ($\phi_{i,\ell}^m$) of every penetrant and the membrane at the downstream membrane face ($z=\ell$), which can be obtained by implementing Eq. (5) with $a_{i,\ell}^m$. Thus, the goal of this step is to estimate the mole fractions ($x_{i,\ell}^s$) and activities ($a_{i,\ell}^s$) of the penetrants at the support layer side and then to estimate the volume fraction ($\phi_{i,\ell}^m$) on the permeate side of the active layer. The mole fractions at the support side are related to the partial molar fluxes through the active layer:

$$N_i = x_{i,\ell}^s \sum_{j=1}^n N_j = \frac{N_i^v}{\bar{V}_i} = x_{i,\ell}^s \frac{N_{total}^v}{\sum_1^n x_{j,\ell}^s \bar{V}_j} \tag{10}$$

Substituting Eq. (10) into Eq. (1) and rearranging, the ordinary different equations (ODEs) can be written as:

$$\frac{d(\phi^m)_{1:n}}{dz} = [\Gamma]^{-1}[B](x_{i,\ell}^s \bar{V})\frac{N_{total}^v}{\sum_1^n x_{j,\ell}^s \bar{V}_j} \tag{11}$$

To solve the ODEs, Eq. (11) is integrated with a criterion of $\sum_i^{n+1}(d\phi_i^m/dz)=0$ (since the sum of volume fractions is always to be 1). Here, the composition of the support fluid ($x_{i,\ell}^s$), which is essentially the same as the permeate fluid, and the total flux ($N_{total}^v$) are unknown variables. To find them, the final integration values for the volume fractions at the permeate side of the active layer ($\phi_{i,\ell}^m$ at z = $\ell$) is used to estimate the composition at the support side ($x_{i,\ell}^s$) using Eq. (9). The ODEs solver is iterated until the iteration guess of ($x_{i,\ell}^{s,iterate}$) and ODE solution ($x_{i,\ell}^{s,solution}$) match. The mole fractions must also sum to one which gives the ($n+1$)$^{st}$ equation. Using this process, the composition of the support fluid ($x_{i,\ell}^s$) and the total flux ($N_{total}^v$) can found.

To quantify [B] (defined by Eq. (2)), the Fickian diffusivities of every molecule permeating though the membrane (Đ$_i^{v,m}$) are first generated from the developed ML diffusion model. Then, they are converted to the Maxwell-Stefan diffusivities (Đ$_i^{v,m}$) by thermodynamically correcting the Fickian diffusivity (to alleviate the loading dependence such that a constant Đ$_i^{v,m}$ could be reasonably assumed):

$$Đ_i^{v,m} = \frac{D_i^{v,m}}{\Gamma_i}; \Gamma_i = \frac{\partial \ln(f_i^m)}{\partial \ln(\phi_i^m)} = \frac{\partial \ln(a_i^m)}{\partial \ln(\phi_i^m)} \tag{12}$$

To complete the diffusion matrix [B], the diffusivity of the mixture can be calculated by averaging diffusion coefficients of all molecules that move through the polymer membrane (Eq. (13)). To separate mixtures primarily consisting of small molecules, glassy polymers have been applied more often than rubbery polymers because glassy polymers are more rigid, such that higher diffusion selectivity is imparted. Despite their increased rigidity, the loss of diffusion selectivity has been observed in glassy polymers that strongly dilate and plasticize in the presence of condensable adsorbates. Less effectiveness in selectivity potentially derives from either a strong coupling between guest species or substantial dilation of the polymer such that guest molecules move as a cohort. In the cohort diffusion case, all molecules in

the mixture still follow individual concentration gradients but have the same effective diffusivity, which is similar to a previously described "sorp-vection" concept[32]. Among various potential averaging methods, this work deployed a volume-corrected interpolation formula of the pure component's Maxwell-Stefan diffusivities.

$$\DJ_i^{v,m} = (\bar{V}_i)^{-1} \prod_{k=1}^{n} (\DJ_k^{v,m} \bar{V}_k)^{\frac{\phi_k^m}{\sum_{j=1}^{n} \phi_j^m}} \quad (13)$$

In summary, to solve the Maxwell-Stefan equations, there are two boundary conditions that must be applied. The initial information only includes chemical structures of the polymer membrane and the permeating molecules and the composition of the feed. Under a given set of operating conditions (e.g., pressure, temperature, composition of the feed mixtures), all the quantities needed for the transport modeling (i.e., Maxwell-Stefan diffusivities, Flory-Huggins interaction parameters, solvent-solvent interaction parameters) are parameterized as input parameters to the transport modeling steps. Note that all diffusivities and sorption uptakes were predicted from the two ML models developed and other solvent properties (e.g., saturation vapor pressure, Hansen solubility parameters, molar volumes and densities of the solvents) were from accessible information (i.e., chemical books, literatures, publicly open chemical information). For solvents that are not available in those sources, the chemical and physical properties (e.g., Hansen solubility parameters, vapor pressure, densities) of the solvents were estimated from ExxonMobil Technology and Engineering Company proprietary correlations which predict these properties based on molecular structure.

### Database for the development of machine learning models
In this work, we used experimentally collected diffusion coefficients and sorption uptake of organic solvents to construct data-driven prediction models for diffusivity and solubility. First, simplified molecular-input line-entry system (so-called SMILES) has been used as the input to describe the polymers and organic solvents. The temperature range for database entries was fixed between 25 °C and 40 °C, which is the temperature at which most of the data in the literature was taken. The thermodynamic activity, which is normally expressed as vapor pressure over saturation vapor pressure at a given temperature, is used as an input feature instead of concentration of solvents in polymers (e.g., mass fraction, volume fraction). In addition, polymers that are post cross-linked or that have high crystallinity from regularly oriented structures were excluded. The dataset includes 2045 diffusivity data points associated with 73 polymers and 151 solvents, and 2275 uptake sorption data points of 46 polymers and 91 solvents (Supplementary Table 1). The polymers consist of homo- and copolymers, and the solvents contain polar/nonpolar and linear/aromatic molecules. Average values were utilized to train the ML model for cases with multiple reported values. Additionally, given the wide range of experimental values, the logarithm 10 of the target property was applied in the model training process.

### Development of ML models (diffusion and sorption)
As shown in Supplementary Fig. 2a, a traditional neural network (NN) model was built for the sorption prediction. This NN model consists of an input layer, two hidden layers, and a final layer for target property prediction (denoted by $\log_{10} S$ where $S$ is sorption uptake, mmol g$^{-1}$). The input layer includes $\log_{10} \hat{V}$ (molar volume of solvents) and $F_n$ (a representation of experimental activities and chemical features of polymers and solvents generated by the hierarchical polymer and molecular fingerprint). The details of the features are summarized in Supplementary Table 1.

In the case of diffusion, we developed a physics-informed ML model to learn the following physical relationship (Supplementary Figure 2b) of $\log_{10} D = A \cdot \log_{10} \hat{V} + B$. Here, $D$ refers to Fickian diffusion

coefficients (cm$^2$ s$^{-1}$) and $\hat{V}$ is the molar volume of solvents. An additional output layer was introduced to predict $A$ and $B$ parameters using $F_n$ features. The output layer is followed by estimating $\log_{10} D$ using $\log_{10} \hat{V}$ and the physical equation above to enforce the NN models to learn the physical relationship.

In both models, the loss function is determined by the root mean square error (RMSE) of the target property ($\log_{10} Y$). To establish the optimized NN models for both properties, we fine-tuned the hyperparameters using KerasTuner (https://keras.io/keras_tuner/) – an automated hyperparameter tuning package. Different number of neurons of the hidden layers, activation functions, dropout ratios, and learning rates were optimized. Also, we adopted 10-fold cross-validation (CV) and the dropout function to avoid overfitting. CV is a common way to validate the generality of developed models by using a portion of validation dataset that is not applied to train the model. In this work, an ensemble of 10 CV models was utilized to provide average and standard deviation of predicted values, given the small dataset. Additionally, the learning curve that describes the RMSE variation of training and test sets as a function of different training set sizes was used to evaluate the model performance. Supplementary Table 1 lists the hyperparameters of final diffusion and sorption prediction models that were trained using the whole dataset and 10-fold CV. All NN models were built using the TensorFlow package.

### Error evaluation
In this work, we applied three types of error metrics, averaged order of magnitude error (AOME), root mean square percentage error (RMSPE), and root mean square error (RMSE), to evaluate the model performance as follows:

$$\text{AOME} = \sum_{1}^{N} \frac{\left| \log_{10} y_{true} - \log_{10} y_{predicted} \right|}{N} \quad (14)$$

$$\text{RMSPE,\%} = \left[ \sum_{1}^{N} \frac{\left( \frac{\log_{10} y_{true} - \log_{10} y_{predicted}}{\log_{10} y_{true}} \right)^2}{N} \right]^{1/2} \cdot 100 \quad (15)$$

$$\text{RMSE} = \left[ \sum_{1}^{N} \frac{\left( \log_{10} y_{true} - \log_{10} y_{predicted} \right)^2}{N} \right]^{1/2} \quad (16)$$

In the error calculations, the scaling in logarithm10 for the true and predicted values was applied to avoid biased error from linear scaling. In addition, the separation factor of a specific component (e.g., guaiacol) was used to evaluate the model performance on a biofuel-type binary mixture separation (e.g., binary mixture of methanol and guaiacol).

$$\text{Separation factor} = \frac{\left( \frac{C_{guaiacol}}{C_{methanol}} \right)_{feed}}{\left( \frac{C_{guaiacol}}{C_{methanol}} \right)_{permeate}} \quad (17)$$

## Data availability
All data are available in the main text or the supplementary materials. Source data are provided with this paper. The needed chemistry-based input parameters for the transport code may be obtained from https://www.polymergenome.org. Source data are provided with this paper.

## Code availability
The code for the mass transport model used in this work may be found at https://github.com/transport-modeling/asyMemSim[38].

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

## Acknowledgements

This work was supported by the ExxonMobil Technology and Engineering Company. Y.J.L. acknowledges the fellowship from Kwanjeong Educational Foundation (South Korea). D.J.W. acknowledges U.S. Department of Energy's Office of Energy Efficiency and Renewable Energy (EERE) under the Advanced Manufacturing Office (award no. DE-EE0007888).

## Author contributions

Y.J.L. and L.C. contributed equally to this work. Databasing diffusivities and solubilities for ML models, support fabrication, coupon-type membrane fabrication by spin-coating, SEM analysis, testing 9-component mixture separations, and predicting all complex mixture separations were conducted by Y.J.L.; Two ML models were constructed by L.C. and the predictions of diffusivities and solubilities were done by J.N.; Fabrication of Torlon membrane and testing 12-component separation were conducted by H.Y.J.; MATLAB code for transport modeling was constructed by D.J.W. and J.K.S.; DUCKY-9 and DUCKY-10 polymers were synthesized by N.C.B.; M.G.F. supervised the polymer synthesis

methods. Crude oil separations, composition analysis, and properties of solvents in the crude oil mixtures were investigated by N.D.R., B.D.M., W.L., and J.R.J.; R.R. and R.P.L. conceived the research. Y.J.L., L.C., R.R., and R.P.L. wrote the paper.

## Competing interests

R.R. is a founder of Matmerize, Inc., a company that provides materials informatics software and services. The remaining authors declare no competing interests.
