## [Peer Review File · Nature Communications]

REVIEWER COMMENTS

Reviewer #1 (Remarks to the Author):

This is a valuable submission worthy of publication if the issues below can be addressed. The use of polymer membranes to achieve liquid phase separations of hydrocarbons has great potential for substantially reducing the energy consumed in the oil and gas industry.

The approach is very specific to the hydrocarbon processing application outlined and my major concern is that the reader may consider its application to other situations where it is less sound. Even in the present case, the fit to the data shown in Figures 4b and d is not exceptionally good. Some of my comments below refer to this issue.

The model is developed based on data available at 25 to 40C, while the tested temperature was 130C. In the present case, the authors argue that this discrepancy is justified as the permeability is not highly temperature dependent. However, this is not usually the case. The authors need to point this out and provide direction as to how temperature effects might be incorporated into the model in the future.

Top of page 5 - The argument that all molecules have the same diffusivity has been examined by the authors in their reference 21. However, I think it needs to be noted here that this argument can only apply to penetrants of comparable nature. The hydrocarbons examined here are all similar, but if water was also present for example, it might not move as part of the cohort. It should also only apply to polymers that are strongly plasticised or swollen. I would not like to see such 'cohort' diffusion applied to low penetrant volume fractions, or in non-plasticised structures. These limitations need to be noted.

The stage cut in experiments is high at 20% and 30%. Was any attempt made to assess the extent of concentration polarisation? This would lead to the accumulation of the larger, less permeable higher boiling point compounds on the membrane surface. Could this explain the higher measured permeance of these compounds for Arab light crude?

The English in the SI needs correction.

To assist the reader, add references for Eq. S1-S5.

SI Page 5 – can you provide some validation for the assumption that 'The activity coefficients (γ_i) of fluid mixtures on the upstream fluid were assumed as unit value'? This seems unlikely to be true.

SI – I note that the authors use a Flory Huggins solubility model, whereas in reference 21 they conclude that a Langmuir-Flory Huggins model is better for glassy polymers. A justification needs to be provided as to why the LM-FH approach is not used in this paper. In particular, competitive sorption is often accounted for within the Langmuir term for glassy polymers and the use of FH for this purpose is not appropriate. Could this also be the reason that the high boiling point components do not fit the model as well?

SI page 7 – clarify what you mean by the 'Support fluid'. Do you mean the fluid in the permeate?

Figure S1 – clarify what is meant by the change in symbol color, including the terminology of Polymer ID.

Reviewer #2 (Remarks to the Author):

The manuscript describes the methodology and results of applying a machine learning driven model for predicting permeation characteristics of multi-component organic mixtures through selective polymeric membranes.

The application is important and I find the contribution to be significant, as it introduces a framework not customarily used in the field, yet, and so can pave the way for others to follow.

In particular, I like the approach that utilizes a physically-sound model framework (the Stefan-Maxwell equations for multi-component diffusion).

The results are well-presented and the manuscript is quite easy to follow. The method well-illustrates its utility and so I believe it should be published and believe it will be a valuable contribution to the literature in the field.

However, there are a few points I wish to raise for the authors to consider as a revision of this paper:

1. The calculations are impressive, but there are some relatively large deviations between predicted and measured fluxes. What are the sources of this? Some discussion is provided but it feels like this is a point worth more thorough examination.

2. Again, with reference to the possible model flaws - I imagine the authors have considered that the assumption of Fickian diffusion might be questionable. Despite the obvious utility of testing the model against a 'real', complex mixture, could it be worthwhile to first validate against a simpler mixture (even simpler than the one used in the preliminary trials, if that makes sense...?), and possibly fine-tune the model parameters? With that in mind, it seems that the sorption/diffusion coefficients are well-represented, so perhaps other parameters could benefit from such 'training'?

3. Curiously, in the mixture separation trials, the largest discrepancies seem to manifest at the highest and lowest fluxes (though the crude oil predictions at low fluxes are excellent, strangely enough). Do the authors have any possible explanations for this?

Ultimately, the main point here are that in order to truly utilize a data-driven model for actual predictions, it is crucial to assess the ability of the physics (which 'grounds' the ML...) to capture the process well. So any sensible points on where discrepancies may come from is a great point for continued research.

minor comments:

1. consider using 'recovery' instead of 'stage-cut'. with are jargon but I believe the first one is commonly used in the membrane separation field, whereas the second isn't...

2. A few details of the model description are a little hard to follow. For example, som of the notation is awkward in its use of brackets (some square, some circular, presumably to denote vectorial/tensorial quantities?). Some streamlining here would definitely benefit the reader interested in the actual details - which is the reader trying to implement the methodology in a future study...

3. in the manuscript, a 'two-point boundary value problem' is mentioned. what does this mean? there is no second order ODE here, so this terminology is confusing to me. please clarify.

Reviewer #3 (Remarks to the Author):

Even though this paper is well written and surely relevant, I consider it of insufficient impact to allow publication in Nature Communication. The novelty is in my opinion limited and the application range of the model too.

Nevertheless, some comments and suggestions:

- The introduction needs to be enhanced by adding more relevant literature
- Term Solvent resistant nanofiltration (SRNF) should come together with the term organic solvent nanofiltration (OSN) as they are equally used.
- One of the most important challenges of OSN with polymer membranes is swelling which can change membrane performance. How do the authors see the effect of this phenomenon in their predicted model?
- Do authors consider the interaction of solvents with each other as well in their model?

- The authors consider chemical structures of the polymer membrane as the only parameter being used in their predicted model. How do they predict changing the voids of a polymer membrane being imposed to different solvents in their model?

Data-driven predictions of complex mixture permeation in polymer membranes

We thank the referees for their constructive feedback and time spent in reviewing this article. We uniformly implemented the suggestions given by the referees and believe the manuscript has been improved through the revision. Point-by-point responses are shown below.

Overall, the changes can be summarized as follows:

- More detailed descriptions about prediction errors and the potential limitations, future works, and perspectives of the data-driven prediction were thoroughly discussed in the manuscript, as suggested by the referees.
- Additional permeation tests and predictions were conducted to assess the capability of the prediction framework and to validate the cohort diffusion assumption in a different mixture separation case (methanol/guaiacol binary mixture). Activity coefficients of solvents in 9-component hydrocarbon mixture were investigated to validate the use of unit activity coefficients in this work.
- Grammatical errors and unclear descriptions in the main manuscript and the Supplementary Information were rectified.

Formatting Note: Throughout this document, **the referees' comments will be shown in blue**, our responses will be shown in black, and changes incorporated into the Manuscript or Supporting Information will be **highlighted text**.

Reviewer: 1

Comments:

This is a valuable submission worthy of publication if the issues below can be addressed. The use of polymer membranes to achieve liquid phase separations of hydrocarbons has great potential for substantially reducing the energy consumed in the oil and gas industry. The approach is very specific to the hydrocarbon processing application outlined and my major concern is that the reader may consider its application to other situations where it is less sound. Even in the present case, the fit to the data shown in Figures 4b and d is not exceptionally good. Some of my comments below refer to this issue

1. The model is developed based on data available at 25 to 40 C, while the tested temperature was 130 C. In the present case, the authors argue that this discrepancy is justified as the permeability is not highly temperature dependent. However, this is not usually the case. The authors need to point this out and provide direction as to how temperature effects might be incorporated into the model in the future.

We thank the referee for this kind summary of our manuscript and for the first point that was raised here. We entirely agree with the comment from the referee. Recent work (ref. 24) on the thermodynamics of ‘crude oil permeation’ showed that the temperature effect has only moderate effects on the separation; however, these moderate effects might not necessarily be true beyond the hydrocarbon permeation systems included here. We do note that the “simple” crude oil experiments (i.e., Figure 3) were conducted at 22 °C, which is well-aligned with the temperatures in the diffusivity and solubility datasets.

To address temperature effects on complex mixture permeation, information of the activation energy of diffusion and the heat of sorption of individual guest molecule should be incorporated into the prediction framework, and this information could potentially be provided by another data-driven model or a simulation (i.e., molecular dynamics). We believe this is crucial future work and this point is added to the manuscript.

- In the revised main manuscript, page 11-12
 - The relatively moderate temperature dependencies observed in the hydrocarbon permeation systems here may not necessarily apply to other systems, which is a current shortcoming of the model. To address this lack of built-in temperature dependence in the model, it will likely be beneficial to develop separate predictors for estimating the activation energy of diffusion and the heat of sorption for individual guest molecules and incorporating these parameters into the transport model. Estimation of these parameters can potentially be achieved through data-based methods and simulations such as molecular dynamics.

2. The argument that all molecules have the same diffusivity has been examined by the authors in their reference 21. However, I think it needs to be noted here that this argument can only apply to penetrants of comparable nature. The hydrocarbons examined here are all similar, but if water was also present for example, the hydrocarbons examined here are all similar, but if water was also present for example, it might not move as part of the cohort. It should also only apply to polymers that are strongly plasticised or swollen. I would not like to see such ‘cohort’ diffusion applied to low penetrant volume fractions, or in non-plasticised structures. These limitations need to be noted.

We thank the referee for the points that are raised here. This is a great point. We also agree with the comment from the referee. We believe that membrane dilation and plasticization in the presence of organic solvents is the main reason behind the apparent lack of diffusion selectivity (which is equivalent to the cohort diffusion case in the manuscript). However, very large molecules (e.g., high boilers in crude oil) likely have reduced diffusivities relative to the average diffusivity exhibited by the smaller molecules; we speculate that the deviations between experimental observations and predictions for high boilers in crude oil in SBAD-1 might derive from this. Moreover, as the referee mentioned, low chemical affinity between dissimilar molecules might also break the cohort diffusion assumption. We have added further clarity in our manuscript regarding potential limitations of the cohort diffusion assumption, as well as potential means in which these limitations can be addressed. **Moreover, we have added a new experimental result to assess the validity of the cohort diffusion case using a biofuel-style separation (methanol and guaiacol).**

The following changes have been made to the manuscript:

- In the revised main manuscript, page 13
 - Additionally, it is important to note that very large molecules (e.g., high boilers in crude oil) likely have reduced diffusivities relative to the average diffusivity exhibited by the smaller molecules.
- In the revised main manuscript, page 12
 - Additionally, in the presence of dissimilar molecules with low chemical affinity, the cohort diffusion assumption might be deemed invalid.
- In the revised main manuscript, page 9-10
 - To further evaluate the prediction capability of the data-driven approach beyond hydrocarbon mixtures, we conducted a separation of a binary mixture consisting of methanol and guaiacol, as a biofuel-type mixture (**Supplementary Fig. 7** and **Supplementary Tables 11-13**). The separation experiment was conducted using three DUCKY-9 membranes. As before, the ML dataset had not been previously trained with these specific solvent molecules for DUCKY-9. Notably, the separation factor of guaiacol was accurately predicted, and the flux was closely predicted to the experimental measurements. This outcome suggests that the data-driven permeation predictions may have broader applicability beyond simple hydrocarbons, which have been the focus of this work, although our limited set of experiments prevents further generalization of this conclusion.
- In the revised Supplemental Information, page 17

Fig. S7.

Separation of a binary mixture of oxygenated molecules as a bio-fuel type mixture via DUCKY-9 membrane. The separation factors of guaiacol and total fluxes are indicated by bars and red circles, respectively. The separation factors were calculated via equation S17. The feed mixture in this crossflow permeation experiment was 80 mol% of methanol and 20 mol% of guaiacol. The test was conducted at 40 bar upstream pressure and 22 °C. The diffusivities and solubilities were predicted by the ML models, and the experimental operating conditions were applied in the transport modeling as described in the method section in this Supplementary Information. The model estimates and solvent properties are summarized in Supplementary Tables S11-S13. A 300 nm membrane thickness was determined from an SEM image of the tested DUCKY-9 membrane (inset). The error bars in concentrations resulted from the deviations in sorption prediction by ML sorption model, and the uncertainty in flux predictions resulted from the deviations in diffusion prediction by ML diffusion model. SR-POLAR model in ASPEN plus was used in the activity coefficient calculation for the non-ideal mixture-based predictions.

- In the revised Supplemental Information, page 28

Table. S11.

Feed concentrations and solvent properties of methanol and guaiacol. δ_D , δ_P , and δ_H are Hansen solubility parameters for dispersion, polarity, and hydrogen-bonding each.

	Hansen solubility parameter (MPa ^{0.5})			Vapor pressure (torr)	Molar volume (cm ³ /mole)
	δ_D	δ_P	δ_H		
methanol	14.7	12.3	22.3	94	40.46
guaiacol	18	7	12	0.103	111.84

- In the revised Supplemental Information, page 29

Table. S12.

Transport parameters of methanol and guaiacol in DUCKY-9 polymer predicted by the ML models: sorption uptakes at unit activity (mmol/g) and Maxwell-Stefan diffusion coefficients ($\mathcal{D}_i^{v,m}$) by thermodynamically correcting Fickian diffusion coefficients ($D_i^{v,m}$) predicted by the ML models at unit activity (cm²/s) of each component in the DUCKY-9 polymer (equation S12). The lowest, average, and highest value were from the uncertainty of the ML predictions. The Flory-Huggins interaction parameters of single component at unit activity were calculated by eq. S6 and S7.

	Lowest sorption (mmol/g)	Sorption in average (mmol/g)	Highest sorption (mmol/g)	Lowest diffusion ($\mathcal{D}_i^{v,m}$, cm ² /s)	Diffusion in average ($\mathcal{D}_i^{v,m}$, cm ² /s)	Highest diffusion ($\mathcal{D}_i^{v,m}$, cm ² /s)
methanol	7.41	9.26	11.75	$1.02 \cdot 10^{-7}$	$2.39 \cdot 10^{-7}$	$5.61 \cdot 10^{-7}$
guaiacol	1.91	2.36	2.88	$4.58 \cdot 10^{-9}$	$1.32 \cdot 10^{-8}$	$3.81 \cdot 10^{-8}$

- In the revised Supplemental Information, page 30

Table S13.

Concentrations (mole fractions) of the binary biofuel mixture feed and permeates (experimentally measured permeate and predicted permeate) separated by DUCKY-9 TFC membrane. Ideal solution for the unity activity coefficient for all components was assumed (top), and the actual

activity coefficients were estimated using SR-POLAR model to account for the non-ideal behavior of the mixture (bottom).

	Feed	Measured permeate	Std. of measurement	Measured total flux (L/m ² /hr)	Predicted permeate	Std. Of prediction	Predicted total flux (L/m ² /hr)
methanol	0.8	0.822	0.002	4.88 ± 0.4	0.821	8.40·10 ⁻⁵	3.69 ± 2.4
guaiaicol	0.2	0.178	0.002		0.179	8.40·10 ⁻⁵	

	Activity coefficients at feed	Activity coefficient at permeate	Predicted permeate	Std. Of prediction	Predicted total flux (L/m ² /hr)
methanol	0.97	0.98	0.823	2.86·10 ⁻⁴	2.81 ± 1.85
guaiaicol	0.78	0.75	0.177	2.86·10 ⁻⁴	

*Std. indicates standard deviation.

3. The stage cut in experiments is high at 20% and 30%. Was any attempt made to assess the extent of concentration polarisation? This would lead to the accumulation of the larger, less permeable higher boiling point compounds on the membrane surface. Could this explain the higher measured permeance of these compounds for Arab light crude?

This is an important point. We agree with the referee that some of the deviations between the experiment and the model (especially for the high boiling point components) likely derive from concentration polarization effects at high stage cuts. We have now added this as discussion in the manuscript, as well as methods to account for that issue in future models.

- In the revised main manuscript page 12-13
 - The higher flux measured here could be attributed to the batch-type fractionation test with a high stage-cut (e.g., 30%), where the feed concentration may have become polarized during the test. To account for the time-related concentration change in feed during batch-type fractionation systems, an additional step that considers the concentration change over time or stage-cut could potentially be included in transport modeling.

4. The English in the SI needs correction and eq.S1-S5 need references.

We thank the referee for this helpful tip. We added the related reference and also edited the SI thoroughly.

5. can you provide some validation for the assumption that ‘The activity coefficients (γ_i) of fluid mixtures on the upstream fluid were assumed as unit value’? This seems unlikely to be true.

This is an important point for predicting complex mixture permeation. We agree that even the relatively ‘simple’ hydrocarbon mixtures are likely not ideal solutions, but they are likely well-approximated as ideal. More advanced thermodynamic models for the feed and permeate phases will be necessary for more complicated mixtures (e.g., biocrudes) in which the activity coefficients are likely to vary widely.

We have updated our 9 component permeation models to account for solution non-ideality. We ran PC-SAFT model (as a common and universal activity coefficient model) to estimate the

activity coefficient values. Most of them are close to unity except a few of them despite the concentrated nature of the mixture. Thus, we feel that our assumption of ideal solutions is defensible for the hydrocarbon work but note that other solutions will need to account for these non-idealities. We appended this activity coefficient calculations and the prediction results with the actual activity coefficient values in the Supplementary Information.

- In the revised Supplemental Information, page 16

Fig. S6.

Validation of ideal solution assumption in predicting hydrocarbon mixture separations (**a**-Matrimid, **b**-DUCKY-9, **c**-DUCKY-10). Comparison between hydrocarbon mixture separation predictions with ideal solution assumption and with non-ideal solution assumption. All experiments and predictions were performed at 22 °C. The pressures in the plots indicate the applied pressure at upstream side with an atmospheric pressure at downstream. The errors in concentrations resulted from the deviations in sorption prediction by ML sorption model, and the uncertainty in flux predictions resulted from the deviations in diffusion prediction by ML diffusion model (Supplementary Tables 4-6). PC-SAFT in ASPEN plus was used in the activity coefficient calculation (Supplementary Tables S8-10). The use of non-ideal activity coefficients in the prediction resulted in slightly higher flux predictions that exhibited better agreement with the experimental measurements. Furthermore, the predicted permeate concentrations showed improve accuracy compared to the measured permeate concentrations, as demonstrated by lower RMSPE and AOME values (Supplementary Tables S8-10).

- In the Supplementary Information, page 25

Table S8.

Concentrations (mole fractions) of the 9-component mixture feed and permeates (experimentally measured permeate and predicted permeate) separated by Matrimid TFC membrane. Ideal solution for the unity activity coefficient for all components was assumed (top), and the actual activity coefficients were estimated using PC-SAFT to account for the non-ideal behavior of the mixture (bottom). RMSPE and AOME were calculated by eq. S14 and eq. S15, respectively.

	Feed	Measured permeate	Std. of measurement	Predicted permeate	Std. Of prediction	RMSPE (%)	AOME
iso-octane	0.117	0.081	0.005	0.104	0.001	11.0	0.16
n-octane	0.188	0.174	0.003	0.190	0.001		
methylcyclohexane	0.197	0.213	0.004	0.203	0.001		
toluene	0.327	0.376	0.002	0.347	0.001		
decalin	0.089	0.080	0.001	0.081	0.001		
tert-butylbenzene	0.039	0.045	0.002	0.039	$2.97 \cdot 10^{-5}$		
iso-cetane	0.011	0.002	$5.04 \cdot 10^{-4}$	0.006	$4.40 \cdot 10^{-4}$		
1,3,5-triisopropylbenzene	0.014	0.003	$6.52 \cdot 10^{-4}$	0.012	$2.74 \cdot 10^{-4}$		
1-methylnaphthalene	0.016	0.022	0.002	0.017	$5.09 \cdot 10^{-5}$		

	Activity coefficients at feed	Activity coefficient at permeate	Predicted permeate	Std. Of prediction	RMSPE (%)	AOME
iso-octane	1.17	1.18	0.105	0.001	10.4	0.15
n-octane	0.99	1.00	0.184	$7.34 \cdot 10^{-4}$		
methylcyclohexane	1.23	1.25	0.202	$5.73 \cdot 10^{-4}$		
toluene	1.27	1.25	0.354	$9.79 \cdot 10^{-4}$		
decalin	0.99	0.98	0.080	$5.71 \cdot 10^{-4}$		
tert-butylbenzene	1.00	0.98	0.039	$2.97 \cdot 10^{-5}$		
iso-cetane	0.83	0.83	0.006	$4.49 \cdot 10^{-4}$		
1,3,5-triisopropylbenzene	0.87	0.86	0.011	$2.86 \cdot 10^{-4}$		
1-methylnaphthalene	1.80	1.74	0.019	$8.16 \cdot 10^{-5}$		

*Std. indicates standard deviation.

- In the Supplementary Information, page 26

Table S9.

Concentrations (mole fractions) of the 9-component mixture feed and permeates (experimentally measured permeate and predicted permeate) separated by DUCKY-9 TFC membrane. Ideal solution for the unity activity coefficient for all components was assumed (top), and the actual activity coefficients were estimated using PC-SAFT to account for the non-ideal behavior of the mixture (bottom). RMSPE and AOME were calculated by eq. S14 and eq. S15, respectively.

	Feed	Measured permeate	Std. of measurement	Predicted permeate	Std. Of prediction	RMSPE (%)	AOME
iso-octane	0.113	0.097	$1.70 \cdot 10^{-4}$	0.077	$5.65 \cdot 10^{-4}$	4.8	0.05
n-octane	0.170	0.170	$4.02 \cdot 10^{-5}$	0.182	$4.02 \cdot 10^{-4}$		
methylcyclohexane	0.202	0.204	$4.84 \cdot 10^{-4}$	0.214	$5.77 \cdot 10^{-4}$		
toluene	0.325	0.368	$2.09 \cdot 10^{-4}$	0.367	$4.12 \cdot 10^{-4}$		
decalin	0.098	0.085	$3.24 \cdot 10^{-4}$	0.077	$5.60 \cdot 10^{-5}$		
tert-butylbenzene	0.041	0.041	$1.79 \cdot 10^{-4}$	0.041	$1.40 \cdot 10^{-4}$		
iso-cetane	0.014	0.006	$3.98 \cdot 10^{-4}$	0.005	$3.73 \cdot 10^{-4}$		
1,3,5-triisopropylbenzene	0.018	0.009	$3.16 \cdot 10^{-4}$	0.013	$2.92 \cdot 10^{-4}$		
1-methylnaphthalene	0.018	0.02	$6.65 \cdot 10^{-5}$	0.021	$3.37 \cdot 10^{-5}$		

	Activity coefficients at feed	Activity coefficient at permeate	Predicted permeate	Std. Of prediction	RMSPE (%)	AOME
iso-octane	1.17	1.19	0.081	$6.94 \cdot 10^{-4}$	5.5	0.05
n-octane	0.99	1.01	0.194	$4.89 \cdot 10^{-4}$		
methylcyclohexane	1.23	1.28	0.203	$4.49 \cdot 10^{-4}$		
toluene	1.27	1.23	0.381	$4.89 \cdot 10^{-4}$		
decalin	0.99	0.98	0.068	$1.63 \cdot 10^{-4}$		
tert-butylbenzene	1.00	0.87	0.039	$1.24 \cdot 10^{-5}$		
iso-cetane	0.83	0.84	0.004	$2.86 \cdot 10^{-4}$		
1,3,5-triisopropylbenzene	0.87	0.85	0.009	$2.44 \cdot 10^{-4}$		
1-methylnaphthalene	1.80	1.67	0.021	$4.71 \cdot 10^{-5}$		

*Std. indicates standard deviation.

- In the Supplementary Information, page 27

Table S10.

Concentrations (mole fractions) of the 9-component mixture feed and permeates (experimentally measured permeate and predicted permeate) separated by DUCKY-10 TFC membrane. Ideal solution for the unity activity coefficient for all components was assumed (top), and the actual activity coefficients were estimated using PC-SAFT to account for the non-ideal behavior of the mixture (bottom). RMSPE and AOME were calculated by eq. S14 and eq. S15, respectively.

	Feed	Measured permeate	Std. of measurement	Predicted permeate	Std. Of prediction	RMSPE (%)	AOME
iso-octane	0.117	0.107	0.003	0.080	0.001	5.9	0.06
n-octane	0.187	0.185	$5.35 \cdot 10^{-4}$	0.197	$5.33 \cdot 10^{-4}$		
methylcyclohexane	0.194	0.196	$2.27 \cdot 10^{-5}$	0.205	$6.99 \cdot 10^{-4}$		
toluene	0.336	0.359	0.007	0.377	$3.89 \cdot 10^{-4}$		
decalin	0.087	0.082	0.002	0.068	$2.68 \cdot 10^{-4}$		
tert-butylbenzene	0.037	0.038	$2.40 \cdot 10^{-4}$	0.039	$4.12 \cdot 10^{-5}$		
iso-cetane	0.011	0.006	$6.94 \cdot 10^{-4}$	0.004	$3.32 \cdot 10^{-4}$		
1,3,5-triisopropylbenzene	0.014	0.009	$8.52 \cdot 10^{-4}$	0.010	$2.57 \cdot 10^{-4}$		
1-methylnaphthalene	0.016	0.017	$1.73 \cdot 10^{-4}$	0.018	$5.27 \cdot 10^{-5}$		

	Activity coefficients at feed	Activity coefficient at permeate	Predicted permeate	Std. Of prediction	RMSPE (%)	AOME
iso-octane	1.17	1.19	0.084	0.001	5.7	0.06
n-octane	0.99	1.01	0.192	$6.12 \cdot 10^{-4}$		
methylcyclohexane	1.23	1.28	0.203	$5.71 \cdot 10^{-4}$		
toluene	1.27	1.24	0.378	$8.17 \cdot 10^{-4}$		
decalin	0.99	0.98	0.069	$3.68 \cdot 10^{-4}$		
tert-butylbenzene	1.00	0.97	0.040	$4.71 \cdot 10^{-5}$		
iso-cetane	0.83	0.84	0.004	$3.26 \cdot 10^{-4}$		
1,3,5-triisopropylbenzene	0.87	0.85	0.009	$2.86 \cdot 10^{-4}$		
1-methylnaphthalene	1.80	1.67	0.021	$9.43 \cdot 10^{-5}$		

*Std. indicates standard deviation.

Nevertheless, the reviewer's comment is still valid with a concern from other mixture cases such as water-contained organic mixtures and polar/nonpolar mixture. Considering this, we have updated the 'limitations' section of the conclusions and highlight the need for incorporation of more sophisticated thermodynamic models in the conclusions.

- In the Supplementary Information, page 5
 - The activity coefficients of hydrocarbons in a 9-component mixture were investigated using the PC-SAFT thermodynamic activity coefficient model in ASPEN Plus. To apply the activity coefficient model to the transport simulation, the phase equilibrium expressions (equation S4 and S9) are updated with values of the estimated activity coefficients each iteration for the downstream phase equilibrium. The upstream equilibrium is fixed at a given temperature, pressure, and composition. The result, presented in Supplementary Tables S8-10, reveals that most of the activity coefficients are nearly unity, despite the concentrated nature of the mixture. However, it is important to note that this ideal mixture assumption may not be applicable to other complex mixtures, particularly those containing water or a combination of polar and nonpolar components. In such cases, more sophisticated thermodynamic models for

both the feed and permeate phases will be necessary to accurately account for the wide variation in activity coefficients.

6. I note that the authors use a Flory Huggins solubility model, whereas in reference 21 they conclude that a Langmuir-Flory Huggins model is better for glassy polymers. A justification needs to be provided as to why the LM-FH approach is not used in this paper. In particular, competitive sorption is often accounted for within the Langmuir term for glassy polymers and the use of FH for this purpose is not appropriate. Could this also be the reason that the high boiling point components do not fit the model as well?

We thank the referee for this point. Langmuir+Flory-Huggins (LM-FH) requires two-parameter isotherm fitting (i.e., for the Langmuir part and the FH part), while FH and competitive FH isotherms can be developed with only one parameter (Flory-Huggins parameter noted as χ_{im} in Eq. S5). To streamline our machine learning predictions, we utilize a one parameter isotherm model (FH), while acknowledging that a two-parameter isotherm would be more accurate yet difficult to accurately predict with our current data set (the dataset is comprised of many ‘unit activity’ swelling experiments and a smaller subset of full isotherms). We have added this point to the discussion in the Supplementary Information.

- In the revised Supplementary Information, page 6-7
 - Another solubility model proposed in a previous study to describe the solubility of a solvent in a polymer consists of two distinct components: Langmuir-type filling of microvoids and Flory-Huggins swelling-type sorption.⁷ However, this model requires fitting two-parameter isotherms for both the Langmuir and Flory-Huggins components. In contrast, the current study employs the Flory-Huggins and competitive Flory-Huggins models, which can be developed with only one parameter, the Flory-Huggins parameter denoted as χ_{im} in Eq. S7. While the two-parameter isotherm would likely be more accurate, the one-parameter Flory-Huggins models were utilized in this work to streamline the predictions of sorption uptakes at unit activity from the ML models.

We also believe that Flory-Huggins (FH) is a useful model to describe sorption of organic liquids/vapors in a polymer system, even in glassy polymer systems, as the sorption of the organic solvent can reduce the glass transition temperature of the polymer sufficiently that FH-type sorption behavior is observed. Prior reports showcase that the FH sorption model can work well for glassy polymers.

- i. A R. Berens, sorption of organic liquids and vapors by rigid PVC, Journal of Applied Polymer Science, 1989, 37, 901-913)
- ii. Leibler et al., On the sorption of Gases and Liquids, in Glassy Polymers, Macromolecules, 1993, 26, 6937-6939)
- iii. Miranda et al., Organic vapor sorption and transport in a thermotropic liquid crystalline polyester, Journal of Membrane Science, 1994, 94, 67-83

- iv. Russell et al., Vapor sorption in plasma polymerized vinyl acetate and methyl methacrylate thin films, *Polymer*, 2001, 42, 2827-2836
- v. Lakshmanan et al., Sorption and transport of organic vapors in poly[1-(trimethylsilyl)-1-propyne], *Journal of Membrane Science*, 1990, 48, 321-331

We added this description and the references above.

- In the revised supplementary information, page 7
 - The Flory-Huggins model is still a useful tool for describing the sorption of organic liquids or vapors in polymer systems, even those that are glassy in nature. This is due to the fact that the sorption of organic solvents can decrease the glass transition temperature of the polymer to a point where Flory-Huggins-type sorption behavior is observed.⁸⁻¹²
- In the revised supplementary information, page 33
 - 8. Berens, A. R. Sorption of organic liquids and vapors by rigid PVC. *Journal of applied polymer science* **37**, 901-913 (1989).
 - 9. Leibler, L. & Sekimoto, K. On the sorption of gases and liquids in glassy polymers. *Macromolecules* **26**, 6937-6939 (1993).
 - 10. Miranda, N., Willits, J., Freeman, B. & Hopfenberg, H. Organic vapor sorption and transport in a thermotropic liquid crystalline polyester. *Journal of membrane science* **94**, 67-83 (1994).
 - 11. Russell, S. & Weinkauff, D. Vapor sorption in plasma polymerized vinyl acetate and methyl methacrylate thin films. *Polymer* **42**, 2827-2836 (2001).
 - 12. Witchey-Lakshmanan, L., Hopfenberg, H. & Chern, R. Sorption and transport of organic vapors in poly [1-(trimethylsilyl)-1-propyne]. *Journal of membrane science* **48**, 321-331 (1990).

7. Clarify what you mean by the ‘support fluid’ and what is meant by the change in symbol color, including the terminology of polymer ID?

Yes. The composition of ‘support fluid’ is essentially same with that of the permeate fluid. This was described in a somewhat confusing way in the original manuscript. We have now clarified this point.

- In the revised Supplemental Information, page 7
 - Here, the composition of the support fluid ($x_{i,\ell}^S$), which is the same as the permeate fluid, and the total flux (N_{total}^V) are unknown variables.

In the Supplementary Figure S1, every polymer ID represents a unique polymer in the dataset, and the IDs are assigned arbitrarily. We included this information in the Supplementary Figure S1.

- In the revised Supplemental Information, page 11
 - **Fig. S1.** Overview of the database that has been used to make ML models: (a) Fickian diffusion coefficients (cm^2/s) and (b) sorption uptakes (mmol/g). Thermodynamic activities (vapor pressure over saturation vapor pressure at a given temperature and

unit activity in cases of liquid sorption) are described by circle sizes in the plots. Every polymer ID represents a unique polymer in the dataset, and the IDs are assigned arbitrarily.

Reviewer #2:

Comments:

The manuscript describes the methodology and results of applying a machine learning driven model for predicting permeation characteristics of multi-component organic mixtures through selective polymeric membranes. The application is important and I find the contribution to be significant, as it introduces a framework not customarily used in the field, yet, and so can pave the way for others to follow. In particular, I like the approach that utilizes a physically-sound model framework (the Stephan-Maxwell equations for multi-component diffusion). The results are well-presented, and the manuscript is quite easy to follow. The method well-illustrates its utility and so I believe it should be published and believe it will be a valuable contribution to the literature in the field. However, there are a few points I wish to raise for the authors to consider as a revision of this paper.

1. The calculations are impressive, but there are some relatively large deviations between predicted and measured fluxes. What are the sources of this? some discussion is provided but it feels like this is a point worth more thorough examination.

We thank the referee for this point. In general, we find that the fluxes are within factors of 1.3-5x of the experimental fluxes – considering the complexity of the permeation system, we are pleased with this level of accuracy, but acknowledge that it can be improved. Importantly, in the 9 component separation experiments, we find that the model correctly “orders” the polymers in terms of their fluxes for that separation.

That said, we agree that additional discussion on the flux deviations should be incorporated in the article. The errors in estimating the diffusivity are in general larger and more impactful than those for sorption on the estimated fluxes. The datasets represent diffusivities in a range of conditions, but often at unit activity for the solvent. In the complex mixture separations, the individual activity of the various compounds in the membrane will not be unity. Moreover, since there is a mixture of solvents in the real experiments, the polymer will be in a state of dilation that is distinct from the state of dilation in the unit activity diffusion experiments. We have considered this effect by changing the free volume in one previous study (reference 21); however, we did not use that approach in this work due to the simplicity of the use of average diffusivity approach. We now make reference to this effect in the manuscript as a potential source of error.

- In the revised main manuscript, page 9
 - An important observation of the predictions made is that the data-driven approach was able to correctly order the polymers based on their respective fluxes for the given separation, even though the predicted fluxes were under-estimated when compared to the measured values. This discrepancy in the predicted and measured fluxes may have resulted from discrepancies in the diffusivity estimates, which were generally larger

and more impactful than the sorption estimates used for predicting the fluxes. The datasets represent diffusivities in a range of conditions but often at unit activity for the solvents. In complex mixture separations, the individual activity of the various compounds in the membrane will not be unity. Additionally, since a mixture of solvents is present in real experiments, the polymer will be in a state of dilation that is distinct from the state of dilation in unit activity diffusion experiments where diffusivity of a pure solvent was determined. A theory related to this effect was envisioned by introducing the free volume change of a polymer membrane when exposed to a complex mixture.²¹ However, this additional complexity was eschewed here due to acceptable prediction accuracy found using the significantly simpler average diffusivity approach.

2. Again, with reference to the possible model flaws - I imagine the authors have considered that the assumption of Fickian diffusion might be questionable. Despite the obvious utility of testing the model against a 'real', complex mixture, could it be worthwhile to first validate against a simpler mixture (even simpler than the one used in the preliminary trials, if that makes sense...?), and possibly fine-tune the model parameters? With that in mind, it seems that the sorption/diffusion coefficients are well-represented, so perhaps other parameters could benefit from such 'training'?

We appreciate this terrific point. It is worth noting that we are utilizing the 'thermodynamically corrected' diffusivity, not the Fickian transport diffusivity. We apologize for not being more clear on this point in the manuscript, and have updated the text to point this out and the tables (Supplementary Table S3 – S6) in the supplementary information with the thermodynamically corrected diffusivities

- In the revised Supplementary Information, page 5
 - $D_{i,n+1}^{v,m}$ is the volume-based Maxwell-Stefan diffusivity of single component i (which is thermodynamically corrected diffusivity by equation 12)

We took the referee's advice and conducted a simpler experiment using oxygenated molecules to evaluate the model performance for non-hydrocarbon binary mixtures. The experimental results are shown below and have been added to the SI along with a comment in the manuscript.

- In the revised main manuscript, page 9
 - To further evaluate the prediction capability of the data-driven approach beyond hydrocarbon mixtures, we conducted a separation of a binary mixture consisting of methanol and guaiacol, as a biofuel-type mixture (**Supplementary Fig. 7** and **Supplementary Tables 11-13**). The separation experiment was conducted using three DUCKY-9 membranes. As before, the ML dataset had not been previously trained with these specific solvent molecules for DUCKY-9. Notably, the separation factor of guaiacol was accurately predicted, and the flux was closely predicted to the experimental measurements. This outcome suggests that the data-driven permeation predictions may have broader applicability beyond simple hydrocarbons, which have been the focus of this work, although our limited set of experiments prevents further generalization of this conclusion.

- In the revised Supplemental Information, page 17

Fig. S7.

Separation of a binary mixture of oxygenated molecules as a bio-fuel type mixture via DUCKY-9 membrane. The separation factors of guaiacol and total fluxes are indicated by bars and red circles, respectively. The separation factors were calculated via equation S17. The feed mixture in this crossflow permeation experiment was 80 mol% of methanol and 20 mol% of guaiacol. The test was conducted at 40 bar upstream pressure and 22 °C. The diffusivities and solubilities were predicted by the ML models, and the experimental operating conditions were applied in the transport modeling as described in the method section in this Supplementary Information. The model estimates and solvent properties are summarized in Supplementary Tables S11-S13. A 300 nm membrane thickness was determined from an SEM image of the tested DUCKY-9 membrane (inset). The error bars in concentrations resulted from the deviations in sorption prediction by ML sorption model, and the uncertainty in flux predictions resulted from the deviations in diffusion prediction by ML diffusion model. SR-POLAR model in ASPEN plus was used in the activity coefficient calculation for the non-ideal mixture-based predictions.

- In the revised Supplemental Information, page 28

Table. S11.

Feed concentrations and solvent properties of methanol and guaiacol. δ_D , δ_P , and δ_H are Hansen solubility parameters for dispersion, polarity, and hydrogen-bonding each.

	Hansen solubility parameter (MPa ^{0.5})			Vapor pressure (torr)	Molar volume (cm ³ /mole)
	δ_D	δ_P	δ_H		
methanol	14.7	12.3	22.3	94	40.46
guaiacol	18	7	12	0.103	111.84

- In the revised Supplemental Information, page 29

Table. S12.

Transport parameters of methanol and guaiacol in DUCKY-9 polymer predicted by the ML models: sorption uptakes at unit activity (mmol/g) and Maxwell-Stefan diffusion coefficients ($\mathcal{D}_i^{v,m}$) by thermodynamically correcting Fickian diffusion coefficients ($D_i^{v,m}$) predicted by the ML models at unit activity (cm^2/s) of each component in the DUCKY-9 polymer (equation S12). The lowest, average, and highest value were from the uncertainty of the ML predictions. The Flory-Huggins interaction parameters of single component at unit activity were calculated by eq. S6 and S7.

	Lowest sorption (mmol/g)	Sorption in average (mmol/g)	Highest sorption (mmol/g)	Lowest diffusion ($\mathcal{D}_i^{v,m}$, cm^2/s)	Diffusion in average ($\mathcal{D}_i^{v,m}$, cm^2/s)	Highest diffusion ($\mathcal{D}_i^{v,m}$, cm^2/s)
methanol	7.41	9.26	11.75	$1.02 \cdot 10^{-7}$	$2.39 \cdot 10^{-7}$	$5.61 \cdot 10^{-7}$
guaiacol	1.91	2.36	2.88	$4.58 \cdot 10^{-9}$	$1.32 \cdot 10^{-8}$	$3.81 \cdot 10^{-8}$

- In the revised Supplemental Information, page 30

Table S13.

Concentrations (mole fractions) of the binary biofuel mixture feed and permeates (experimentally measured permeate and predicted permeate) separated by DUCKY-9 TFC membrane. Ideal solution for the unity activity coefficient for all components was assumed (top), and the actual activity coefficients were estimated using SR-POLAR model to account for the non-ideal behavior of the mixture (bottom).

	Feed	Measured permeate	Std. of measurement	Measured total flux ($\text{L}/\text{m}^2/\text{hr}$)	Predicted permeate	Std. Of prediction	Predicted total flux ($\text{L}/\text{m}^2/\text{hr}$)
methanol	0.8	0.822	0.002	4.88 ± 0.4	0.821	$8.40 \cdot 10^{-5}$	3.69 ± 2.4
guaiacol	0.2	0.178	0.002		0.179	$8.40 \cdot 10^{-5}$	

	Activity coefficients at feed	Activity coefficient at permeate	Predicted permeate	Std. Of prediction	Predicted total flux ($\text{L}/\text{m}^2/\text{hr}$)
methanol	0.97	0.98	0.823	$2.86 \cdot 10^{-4}$	2.81 ± 1.85
guaiacol	0.78	0.75	0.177	$2.86 \cdot 10^{-4}$	

*Std. indicates standard deviation.

We agree with the referee that other parameters can be incorporated into the transport model to improve the robustness and accuracy of this approach. One parameter could be the concavity/convexity of the sorption isotherm, for instance. We feel that these are important points for future research, and we have added a comment to the manuscript.

- In the revised Supporting Information, page 7
 - To improve the robustness and accuracy of the data-driven approach, it is possible to envisage the inclusion of other parameters such as the concavity/convexity of an isotherm through the use of additional machine learning algorithms in the future. By integrating these parameters with the existing Flory-Huggins sorption model utilized in this study, the predictive capability of the model could potentially be improved.

3. Curiously, in the mixture separation trials, the largest discrepancies seem to manifest at the highest and lowest fluxes (though the crude oil predictions at low fluxes are excellent, strangely enough). Do the authors have any possible explanations for this?

This is a good observation. We note that the lowest fluxes are often high boiling point components in which we have the sparsest data, so there is the potential that the diffusivities are not as well predicted in that case. Beyond this, the “smallest” and “largest” molecules in the mixture are likely to deviate from the cohort diffusion assumption (it is likely that the diffusivities are on some sort of distribution – most are well-represented by the average, but some are not). We have added a comment to the manuscript based on this important observation from the referee.

- In the revised main manuscript, page 12
 - We note that magnitude of the discrepancies between the model and the experiments are most significant for the components with the highest and lowest fluxes. While the cohort diffusion assumption implies that all penetrants have the same diffusivity within the membrane, in reality the diffusivities are likely to exist on a distribution such that the penetrants farthest away from the mean diffusivity exhibit the largest deviations from the experimental observations.

Ultimately, the main point here is that in order to truly utilize a data-driven model for actual predictions, it is crucial to assess the ability of the physics (which grounds the ML...) to capture the process well. So any sensible points on where discrepancies may come from is a great point for continued research.

We thank the referee for coming up with this important point for the article. We may be able to pull up two major points in general for this point.

First, since solvent permeance is essentially driven by both diffusion and sorption of guest molecules in a polymer membrane in solution-diffusion type permeation regime, accurate predictions of these two parameters is of utmost importance for any transport model to be effective. This is a major distinction in our approach from other ML-based permeability predictors in the literature – we note that a pure component permeability (which is often a reliable predictor of transport rates in gas separations) is a poor predictor of solvent fluxes in mixtures. That said, the solvent solubility and diffusivity in a polymer membrane is dependent on the current state of the polymer (e.g., level of dilation, aging, processing history, etc.), and the mixture permeating through it. Our current model cannot address these issues, but nonetheless provides a surprisingly accurate estimate of the membrane performance even without these important but ultimately complicated ‘real world’ issues. We recommend that future models work to incorporate the physics of the polymer and polymer free volume into the transport framework. We added this potential future work into the conclusion in the main manuscript.

- In the revised main manuscript, page 13-14
 - Accurate estimates of a solvent’s solubility and diffusivity within a polymer are necessary to enable predictions of solvent permeation through polymer membranes. These two parameters depend on the current state of the polymer, including its level of dilation, aging, and processing history, as well as the mixture that permeates through

the membrane. Although our current model can estimate membrane performance with surprising accuracy without taking into account these complicated issues, future models that incorporate the physics of the polymer and its free volume into the transport framework should result in improved model accuracy.

*minor comments:

1. consider using 'recovery' instead of 'stage-cut'.

We note that recovery and stage-cut are slightly different quantities. To avoid confusion, we have explicitly defined stage-cut in the SI.

- In the revised Supplemental Information page 3
 - stage cut is the mass fraction of the feed that permeates through the membrane

2. A few details for the model description are a little hard to follow. For example, some of the notation is awkward in its use of brackets (some square, some circular, presumably to denote vectorial/tensorial quantities?). Some streamlining here would definitely benefit the reader interested in the actual details - which is the reader trying to implement the methodology in a future study...

The circular ones were used to indicate the n -dimensional vector and the brackets were used to indicate the matrix for diffusion and thermodynamic correction matrix. We edited the descriptions of the transport modeling in the Supplementary Information.

- In the revised Supplemental Information, page 5
 - (N_i^v) is an n -dimensional vector of fluxes (L/m²/hr) of permeants, [B] is an ($n \times n$)-dimensional diffusional matrix, [Γ] is an ($n \times n$)-dimensional thermodynamic coupling matrix

3. In the manuscript, a 'two-boundary value problem' is mentioned. What does this mean? There is no second order ODE here, so this terminology is confusing to me. Please clarify.

We meant there are two boundaries (e.g., upstream and downstream) to solve the differential equations, and a '2-point boundary value problem' was noted in the manuscript. However, we agree that this description might be somewhat confusing. We have included a parenthetical brief description.

- In the revised main manuscript, page 7
 - Using this information, a 2-point boundary value problem (i.e., a boundary at the upstream and downstream sides of the membrane) must be solved for the N -component system, which includes the polymer membrane and all of the solvents in the mixture.

Reviewer #3:

Comments:

Even though this paper is well written and surely relevant, I consider it of insufficient impact to

allow publication in Nature Communication. The novelty is in my opinion limited and the application range of the model too.

We thank the referee for giving a critical opinion and spending time to review the paper.

We want to highlight the potential impact and the novelty of this work. The importance of separating various type of complex mixtures such as crude oils has been growing, and polymer membranes have demonstrated promise for these critical separation tasks. Therefore, the ability to predict the performance of existing or new materials is very important in the design and selection of materials for development of new separation processes. Data science approaches such as machine learning have already significantly advanced gas separation membranes to the point that artificial intelligence is capable of drawing new structures of polymers that achieve targeted performance metrics. Recent attempts have been made to predict the performance of OSN/SRNF and OSRO-type polymer membranes, and some of these have used data-based methods. However, these works are narrowly limited in their use because the models were trained with a few commercial membranes and do not have any framework that can parameterize the separation conditions such as activities and pressures. In addition, most of the works were mainly designed to estimate the pure solvent permeance, which we argue is not a useful parameter when we look at a complex mixture separation.

In this regard, our prediction strategy is unique and novel. The integration of ML predictors and mass transport modeling essentially enables the prediction that starts from the accessible information such as structures of the polymers and solvents and also separation operation conditions can be taken into as input parameters in the transport modeling.

Importantly, the ML models can simply be used to predict solvent diffusivities and solubilities in polymers – this has wide applicability beyond membranes. Moreover, our objective for this paper was to predict the separation of any mixture through any linear polymer membrane – this has wide applicability and impact.

1. The introduction needs to be enhanced by adding more relevant literatures

We thank the referee for suggesting this. We think a unique feature of our work is not only generating diffusivities and solubilities of various component in a mixture, but also the ability to account for mixture concentrations and operation conditions. Incorporating these variables into the modeling is essential to capture the sheer variety of applications available to these types of membranes. We highlighted this point in the manuscript (introduction part) and added relevant literatures to enhance the introduction as the referee suggested.

- In the revised main manuscript, page 3
 - Moreover, existing ML and modeling approaches are not generalizable to the sheer variety of solvent molecules nor a wide range of polymer materials under of the large phase space of operating conditions (e.g., feed concentrations, pressures), which is critical for predicting the performance of a membrane.¹⁸⁻²⁰

- In the revised main manuscript, page 16

18. Jang, H. Y. *et al.* Torlon® hollow fiber membranes for organic solvent reverse osmosis separation of complex aromatic hydrocarbon mixtures. *AIChE Journal* **65**, e16757 (2019).

19. Verbeke, R. *et al.* Solutes in solvent resistant and solvent tolerant nanofiltration: How molecular interactions impact membrane rejection. *Journal of Membrane Science*, 121595 (2023).

20. Hesse, L., Mićović, J., Schmidt, P., Górak, A. & Sadowski, G. Modelling of organic-solvent flux through a polyimide membrane. *Journal of membrane science* **428**, 554-561 (2013).

2. Term Solvent-Resistant nanofiltration (SRNF) should come together with the term organic solvent nanofiltration (OSN) as they are equally used.

We have added a parenthetical noting that OSN is sometimes referred to as SRNF.

- In the revised main manuscript, page 3
 - ML approaches have been recently applied to organic solvent nanofiltration (OSN, also referred to as solvent resistant nanofiltration, SRNF) to predict permeance of a single solvent and the rejection of a single solute.^{15, 16}

3. One of the most important challenges of OSN with polymer membranes is swelling which can change membrane performance. How do the authors see the effects of this phenomenon in their predicted model?

This is a great point. Importantly, the solubility and diffusivity data are largely from experiments conducted at unit activity – thus, the experiments have the polymer membrane dilation physics ‘baked into’ the dataset.

One important challenge is that the polymer dilation will depend on the exact composition of the mixture permeating through the membrane and that this will influence the individual diffusivity of each compound permeating through the membrane. We have considered this challenge in prior work, but we ultimately find that the ‘average’ diffusion approach used here is robust.

4. Do authors consider the interaction of solvents with each other as well in their model.

The interaction between solvents is taken into the Flory-Huggins type competitive sorption (Eq. S5 in the original manuscript). In this equation, χ_{ij} indicates the chemical interaction between different solvents, and this is calculated by the equation S8. The closer chemical affinity to each other solvent results in lower values of this solvent-solvent interaction parameters. We have further emphasized this in the manuscript:

- In the revised Supplemental Information, page 6
 - χ_{ij} is the binary solvent-solvent interaction parameter that is calculated using a modified Hansen solubility theory. This accounts for the chemical interaction between the molecules within the membrane

5. The authors consider chemical structures of the polymer membrane as the only parameter being used in their predicted model. How do they predict changing the voids of a polymer membrane being imposed to different solvents in their model?

As noted in the previous comment, our hypothesis is that ‘hidden information’ such as this is intrinsically captured in the experimental data. We agree that deviations in the free volume and levels of dilation will occur between pure solvent experiments and mixture experiments, yet we find that the average diffusion approach is workable despite these deviations. We added this point and described a perspective on future work in the conclusion part.

- In the revised main manuscript, page 13-14
 - Accurate estimates of a solvent’s solubility and diffusivity within a polymer are necessary to enable predictions of solvent permeation through polymer membranes. These two parameters depend on the current state of the polymer, including its level of dilation, aging, and processing history, as well as the mixture that permeates through the membrane. Although our current model can estimate membrane performance with surprising accuracy without taking into account these complicated issues, future models that incorporate the physics of the polymer and its free volume into the transport framework should result in improved model accuracy.

REVIEWERS' COMMENTS

Reviewer #1 (Remarks to the Author):

The authors have addressed all my comments and I am happy for the paper to be accepted.

Reviewer #2 (Remarks to the Author):

I have gone over the revised manuscript and the response letter.

The authors have made all necessary corrections and I recommend its publication.

One minor outstanding comment:

the notion of a two-boundary problem, in relation to a 1st order ODE, remains confusing to me. there can only be one boundary condition for this equation. see how this can be made clearer to the reader.

Reviewer #3 (Remarks to the Author):

I think the authors have taken all remarks and comments very seriously, and responded very well to them.

However, and despite the high quality of this manuscript as such, I still think that impact and novelty are somewhat too limited for such high-IF journal with broad audience as Nature Communications.

Nevertheless, as both other reviewers seem more positive and if the editor also agrees, I can anyhow agree with acceptance of the paper.

Data-driven predictions of complex organic mixture permeation in polymer membranes

We thank the referees for their time spent in reviewing this article and our response to the reviewers' comments. Point-by-point responses are shown below.

Formatting Note: Throughout this document, **the referees' comments will be shown in blue**, our responses will be shown in black, and changes incorporated into the Manuscript or Supporting Information will be **highlighted text**.

Reviewer #1:

The authors have addressed all my comments and I am happy for the paper to be accepted.

We thank the referee for giving very helpful feedback and reviewing all response made in this article.

Reviewer #2:

I have gone over the revised manuscript and the response letter.

The authors have made all necessary corrections and I recommend its publication.

One minor outstanding comment:

the notion of a two-boundary problem, in relation to a 1st order ODE, remains confusing to me. there can only be one boundary condition for this equation. see how this can be made clearer to the reader.

We thank the referee for reviewing all response made in this article and coming up this important point for the article. As the referee suggested, we added more detailed description in the main text to clarify the ODE solver, as below:

While it is true that the ordinary differential equation (ODE) solvent only requires on initial boundary condition at the upstream side of the membrane, the unknown variables of the total flux and permeate composition make the downstream boundary condition an unknown as well. This necessitates an iterative numerical procedure, thus creating the 2-point boundary value problem. (see Methods section for further details).

Reviewer #3:

I think the authors have taken all remarks and comments very seriously, and responded very well to them.

However, and despite the high quality of this manuscript as such, I still think that impact and novelty are somewhat too limited for such high-IF journal with broad audience as Nature Communications.

Nevertheless, as both other reviewers seem more positive and if the editor also agrees, I can anyhow agree with acceptance of the paper.

We thank the referee for reviewing all responses made in this article and for providing constructive feedback. The input ultimately improved the article, for which we are thankful.